# Contextual Active Model Selection

**Xuefeng Liu**[1][*]**, Fangfang Xia**[2]**, Rick L. Stevens**[1,2]**, Yuxin Chen**[1]
[1]Department of Computer Science, University of Chicago
[2]Argonne National Laboratory

## Abstract

While training models and labeling data are resource-intensive, a wealth of pre-trained models and unlabeled data exists. To effectively utilize these resources, we present an approach to actively select pre-trained models while minimizing labeling costs. We frame this as an *online contextual active model selection* problem: At each round, the learner receives an unlabeled data point as a context. The objective is to adaptively select the best model to make a prediction while limiting label requests. To tackle this problem, we propose CAMS, a contextual active model selection algorithm that relies on two novel components: (1) a contextual model selection mechanism, which leverages context information to make informed decisions about which model is likely to perform best for a given context, and (2) an active query component, which strategically chooses when to request labels for data points, minimizing the overall labeling cost. We provide rigorous theoretical analysis for the regret and query complexity under both adversarial and stochastic settings. Furthermore, we demonstrate the effectiveness of our algorithm on a diverse collection of benchmark classification tasks. Notably, CAMS requires substantially less labeling effort (less than 10%) compared to existing methods on CIFAR10 and DRIFT benchmarks, while achieving similar or better accuracy.

## 1 Introduction

As pre-trained models become increasingly prevalent in a variety of real-world machine learning applications [2, 11, 53], there is a growing demand for label-efficient approaches for model selection, especially when facing varying data distributions and contexts at run time. Oftentimes, no single pre-trained model achieves the best performance for every context, and a proper approach is to construct a policy for adaptively selecting models for specific contexts [48]. For instance, in medical diagnosis and drug discovery, accurate predictions are of paramount importance. The diagnosis of diseases through pathologist or the determination of compound chemical properties through lab testing can be costly and time-consuming. Different models may excel in analyzing different types of pathological images [1, 3, 23] or chemical compounds [17, 32, 46]. Furthermore, in many real-world applications, the collection of labels for model evaluation can be expensive and data instances may arrive as a stream rather than all at once. This scenario necessitates *cost-effective* and *robust* online algorithms capable of determining the most efficient model selection policy even when faced with a limited supply of labels, a scenario not fully addressed by previous works that typically assume access to all labels [6, 7, 27, 54].

Recently, the problem of online model selection with the consideration of label acquisition costs was studied in a *context-free* setting by Karimi et al. [39]. However, this approach doesn't fully capture the dynamics of data contexts that are essential in many applications. Recognizing this gap, in this paper, we consider a more general problem setting that incorporates context information for adaptive model selection. We introduce CAMS, an algorithm for active model selection that dynamically adapts to the data context to choose the most suitable models for an arbitrary data stream. As highlighted in

---

[*]Correspondence to: Xuefeng Liu <xuefeng@uchicago.edu>.

38th Conference on Neural Information Processing Systems (NeurIPS 2024).

| Setup \ Algorithm | Online bagging* [54] bagging | Hedge [27] online learning | EXP3 [7] bandit | EXP4 [7] contextual bandits | Query by Committee [65] active learning | ModelPicker [39] model selection | CAMS (ours) (ours) |
|---|---|---|---|---|---|---|---|
| model selection† | × | ✓ | ✓ | ✓ | × | ✓ | ✓ |
| full-information | ✓ | ✓ | × | × | ✓ | ✓ | ✓ |
| active queries | × | × | × | × | ✓ | ✓ | ✓ |
| context-aware | × | × | × | ✓ | × | × | ✓ |

† We regard "arms" as "models" when comparing CAMS against bandit algorithms, such as EXP3/EXP4.
* Online ensemble learning aims to build a composite model by aggregating multiple models rather than selecting the best model (for a given context).

Table 1: Comparing CAMS against related work in terms of problem setup.

Table 1, CAMS aims to address the need for adaptive and effective model selection, by bridging the gap between contextual bandits, online learning, and active learning.

Our key contributions are summarized as follows:

- We investigate a novel problem which we refer to as *contextual active model selection*, and introduce a novel principled algorithm that features two key technical components: (1) a *contextual online model selection* procedure, designed to handle both stochastic and adversarial settings, and (2) an *active query* strategy. The proposed algorithm is designed to be robust to heterogeneous data streams, accommodating both stochastic and adversarial online data streaming scenarios.

- We provide rigorous theoretical analysis on the *regret* and *query complexity* of the proposed algorithms. We establish regret upper bounds for both adversarial and stochastic data streams under limited label costs. Our regret upper bounds are within constant factors of the existing lower bounds for online learning problems with expert advice under the full information setting.

- Empirically, we demonstrate the effectiveness and robustness of our approach on a variety of online model selection tasks spanning different application domains (from generic ML benchmarks such as CIFAR10 to domain-specific tasks in biomedical analysis), data scales (ranging from 80 to 10K), data modalities (i.e., tabular, image, and graph-based data), and label types (binary or multiclass labels). For the tasks evaluated, (1) CAMS outperforms all competing baselines by a significant margin. (2) Asymptotically, CAMS performs no worse than the best single model. (3) CAMS is not only robust to adversarial data streams but also can efficiently recover from "malicious experts" (i.e. inferior pre-trained models).

## 2 Related Work

**Contextual bandits.** Classical bandit algorithms (e.g., [6, 7]) aim to find the best arm(s) through a sequence of actions. When side information (e.g., user profile for recommender systems or environmental context for experimental design) is available, many bandit algorithms can be lifted to the contextual setting: For example, EXP4 [7, 9, 52] considers the bandit setting with expert advice: At each round, experts announce their predictions of which actions are the most promising for the given context, and the goal is to construct a expect selection policy that competes with the best expert from hindsight. In bandit problems, the learner only gets to observe the reward for each action taken. In contrast, for the online model selection problem considered in this work—where an action corresponds to choosing a model to make prediction on an incoming data point—we get to see the loss/reward of *all* models on the labeled data point. By utilizing the information from unchosen arms, it could significantly reduce the cumulative regret. In this regard, this work aligns more closely with online learning with *full information* setting, where the learner has access to the loss of all the arms at each round (e.g. as considered in the Hedge algorithm [13, 14, 27, 36]).

**Online learning with full information.** A clear distinction between our work and online learning is that we assume the labels of the online data stream are not readily available but can be acquired at each round with a cost. In addition, the learner only observes the loss incurred by all models on a data point when it decides to query its label. In contrast, in the canonical online learning setting, labels arrive with the data and one gets to observe the loss of all candidate models at each round. Similar setting also applies to other online learning problems, such as online boosting or bagging. A related work to ours is online learning with label-efficient prediction [16], which proposes an online learning algorithm with matching upper and lower bounds on the regret. However, they consider a fixed query probability that leads to a linear query complexity. Our algorithm, inspired by uncertainty sampling in active learning, achieves an improved query complexity with the adaptive query strategy while maintaining a comparable regret.

**Stream-based Active learning.** Active learning aims to achieve a target learning performance with fewer training examples [64]. The active learning framework closest to our setting is query-by-committee (QBC) [65], in particular under the stream-based setting [35, 47]. QBC maintains a committee of hypotheses; each committee member votes on the label of an instance, and the instances with the maximal disagreement among the committee are considered the most informative labels. Note that existing stream-based QBC algorithms are designed and analyzed assuming i.i.d. data streams. In comparison, our work uses a different query strategy as well as a novel model recommendation strategy, which also applies to the adversarial setting.

**Active model selection.** Active model selection captures a broad class of problems where model evaluations are expensive, either due to (1) the cost of evaluating (or "probing") a model, or (2) the cost of annotating a training example. Existing works under the former setting [49, 59] and online setting [21, 66] often ignore context information and data annotation cost, and only consider *partial* feedback on the models being evaluated/ probed on i.i.d. data. The goal is to identify the best model with as few model probes as possible. This is quite different from our problem setting which considers the full information setting as well as non-negligible data annotation cost. [71] proposes that the optimal model choice is influenced by the sample size rather than any individual sample feature. [44] addresses the active model selection problem, however both works do not adopt a stream-based approach. For the later, apart from Karimi et al. [39], an online contextual-free model selection work, as shown in Table 1, most existing works assume a pool-based setting where the learner can choose among the pool of unlabeled data [4, 29, 43, 49, 60, 61, 68, 76], and the goal is to identify the best model with a minimal set of labels.

## 3 Problem Statement

**Notations.** Let $\mathcal{X}$ be the input domain and $\mathcal{Y} := \{0, \ldots, c-1\}$ be the set of $c$ possible class labels for each input instance. Let $\mathcal{F} = \{f_1, \ldots, f_k\}$ be a set of $k$ pre-trained classifiers over $\mathcal{X} \times \mathcal{Y}$. A model selection policy $\pi : \mathcal{X} \to \Delta^{k-1}$ maps any input instance $\boldsymbol{x} \in \mathcal{X}$ to a distribution over the pre-trained classifiers $\mathcal{F}$, specifying the probability $\pi(\boldsymbol{x})$ of selecting each classifier under input $\boldsymbol{x}$. Here, $\Delta^{k-1}$ denotes the $k$-dimensional probability simplex $\{\boldsymbol{w} \in \mathbb{R}^k : |\boldsymbol{w}| = 1, \boldsymbol{w} \geq 0\}$. One can interpret a policy $\pi$ as an "expert" that suggests which model to select for a given *context* $\boldsymbol{x}$.

Let $\Pi$ be a collection of model selection policies. In this paper, we propose an *extended policy set* $\Pi^* := \Pi \cup \{\pi_1^{\text{const}}, \ldots, \pi_k^{\text{const}}\}$ which also includes constant policies that always suggest a fixed model. Here, $\pi_j^{\text{const}}(\cdot) := \boldsymbol{e}_j$, and $\boldsymbol{e}_j \in \Delta^{k-1}$ denotes the canonical basis vector with $e_j = 1$. Unless otherwise specified, we assume $\Pi$ is finite with $|\Pi| = n$, and $|\Pi^*| \leq n + k$. As a special case, when $\Pi = \emptyset$, our problem reduces to the contextual-free setting.

**The contextual active model selection protocol.** Assume that the learner knows the set of classifiers $\mathcal{F}$ as well as the set of model selection policies $\Pi$. At round $t$, the learner receives a data instance $\boldsymbol{x}_t \in \mathcal{X}$ as the context for the current round, and computes the predicted label $\hat{y}_{t,j} = f_j(\boldsymbol{x}_t)$ for each pre-trained classifier indexed by $j \in [k]$. Denote the vector of predicted labels by all $k$ models by $\hat{\boldsymbol{y}}_t := [\hat{y}_{t,1}, \ldots, \hat{y}_{t,k}]^\top$. Based on previous observations, the learner identifies a model/classifier $f_{j_t}$ and makes a prediction $\hat{y}_{t,j_t}$ for the instance $\boldsymbol{x}_t$. Meanwhile, the learner can obtain the true label $y_t$ *only if* it decides to query $\boldsymbol{x}_t$. Upon observing $y_t$, the learner incurs a *query cost*, and receives a (full) loss vector $\boldsymbol{\ell}_t = \mathbb{I}_{\{\hat{\boldsymbol{y}}_t \neq y_t\}}$, where the $j$th entry $\ell_{t,j} := \mathbb{I}_{\{\hat{y}_{t,j} \neq y_t\}}$ corresponds to the 0-1 loss for model $j \in [k]$ at round $t$. The learner can then use the queried labels to adjust its model selection criterion for future rounds.

**Performance metric.** If $\boldsymbol{x}_t$ is misclassified by the model $j_t$ selected by learner at round $t$, i.e. $\hat{y}_{t,j_t} \neq y_t$, it will be counted towards the *cumulative loss* of the learner, regardless of the learner making a query. Otherwise, no loss will be incurred for that round. For a learning algorithm $\mathcal{A}$, its cumulative loss over $T$ rounds is defined as $L_T^{\mathcal{A}} := \sum_{t=1}^T \ell_{t,j_t}$.

In practice, the choice of model $j_t$ at round $t$ by the learner $\mathcal{A}$ could be random: For *stochastic* data streams where $(\boldsymbol{x}, y)$ arrives i.i.d., the learner may choose different models for different random realizations of $(\boldsymbol{x}_t, y_t)$. For the *adversarial* setting where the data stream $\{(\boldsymbol{x}_t, y_t)\}_{t \geq 1}$ is chosen by an adversary before each round, the learner may randomize its choice of model to avoid a constant loss at each round [33]. Therefore, due to the randomness of $L_T^{\mathcal{A}}$, we consider the *expected* cumulative loss $\mathbb{E}[L_T^{\mathcal{A}}]$ as a key performance measure of the learner $\mathcal{A}$. To characterize the progress of $\mathcal{A}$, we

consider the *regret*—formally defined as follows— as the difference between the cumulative loss received by the learner and the loss if the learner selects the "best policy" $\pi^* \in \Pi^*$ in hindsight.

For stochastic data streams, we assume that each policy $i$ recommends the *most probable* model w.r.t. $\pi_i(\boldsymbol{x}_t)$ for context $\boldsymbol{x}_t$. We use $\mathrm{maxind}(\boldsymbol{w}) := \arg\max_{j:w_j \in \boldsymbol{w}} w_j$ to denote the index of the maximal-value entry[2] of $\boldsymbol{w}$. Since $(\boldsymbol{x}, y)$ are drawn i.i.d., we define $\mu_i = \frac{1}{T}\sum_{t=1}^{T} \mathbb{E}_{\boldsymbol{x}_t, y_t}\left[\ell_{t,\mathrm{maxind}(\pi_i(\boldsymbol{x}_t))}\right]$. This leads to the pseudo-regret for the stochastic setting over $T$ rounds, defined as

$$\overline{\mathcal{R}}_T(\mathcal{A}) = \mathbb{E}[L_T^{\mathcal{A}}] - T \min_{i \in [|\Pi^*|]} \mu_i. \tag{1}$$

In an adversarial setting, since the data stream (and hence the loss vector) is determined by an adversary, we consider the reference best policy to be the one that minimizes the loss on the adversarial data stream, and the expected regret

$$\mathcal{R}_T(\mathcal{A}) = \mathbb{E}[L_T^{\mathcal{A}}] - \min_{i \in [|\Pi^*|]} \sum_{t=1}^{T} \tilde{\ell}_{t,i}, \tag{2}$$

where $\tilde{\ell}_{t,i} := \langle \pi_i(\boldsymbol{x}_t), \boldsymbol{\ell}_t \rangle$ denotes the expected loss if the learner commits to policy $\pi_i$, randomizes and selects $j_t \sim \pi_i(\boldsymbol{x}_t)$ (and receives loss $\ell_{t,j_t}$) at round $t$. Our goal is to devise a principled online active model selection strategy to minimize the regret as defined in (1) or (2), while maintaining a low total query cost. For convenience, we refer the readers to App. B for a summary of the notations used in this paper.

## 4   Contextual Active Model Selection

In this section, we introduce our main algorithm for both stochastic and adversarial data streams.

1: **Input:** Models $\mathcal{F}$, policies $\Pi$, #rounds $T$, budget $b$
2: Initialize loss $\tilde{\boldsymbol{L}}_0 \leftarrow 0$; query cost $C_0 \leftarrow 0$
3: Set $\Pi^* \leftarrow \Pi \cup \{\pi_1^{\mathrm{const}}, \ldots, \pi_k^{\mathrm{const}}\}$ according to Eq. (3)
4: **for** $t = 1, 2, ..., T$ **do**
5:     Receive $\boldsymbol{x}_t$
6:     $\eta_t \leftarrow \mathrm{SETRATE}(t, \boldsymbol{x}_t, |\Pi^*|)$
7:     Set $q_{t,i} \propto \exp\left(-\eta_t \tilde{L}_{t-1,i}\right) \forall i \in |\Pi^*|$
8:     $j_t \leftarrow \mathrm{RECOMMEND}(\boldsymbol{x}_t, \boldsymbol{q}_t)$
9:     Output $\hat{y}_{t,j_t} \sim f_{t,j_t}$ as the prediction for $\boldsymbol{x}_t$
10:     Compute $z_t$ in Eq. (4)
11:     Sample $U_t \sim \mathrm{Ber}(z_t)$
12:     **if** $U_t = 1$ and $C_t \leq b$ **then**
13:         Query the label $y_t$
14:         $C_t \leftarrow C_{t-1} + 1$
15:         Compute $\boldsymbol{\ell}_t$: $\ell_{t,j} = \mathbb{I}\{\hat{y}_{t,j} \neq y_t\}, \forall j \in [|\mathcal{F}|]$
16:         Estimate model loss: $\hat{\ell}_{t,j} = \frac{\ell_{t,j}}{z_t}, \forall j \in [|\mathcal{F}|]$
17:         Update $\tilde{\boldsymbol{\ell}}_t$: $\tilde{\ell}_{t,i} \leftarrow \langle \pi_i(\boldsymbol{x}_t), \hat{\ell}_{t,j} \rangle, \forall i \in [|\Pi^*|]$
18:         $\tilde{\boldsymbol{L}}_t = \tilde{\boldsymbol{L}}_{t-1} + \tilde{\boldsymbol{\ell}}_t$
19:     **else**
20:         $\tilde{\boldsymbol{L}}_t = \tilde{\boldsymbol{L}}_{t-1}$
21:         $C_t \leftarrow C_{t-1}$
22:     **end if**
23: **end for**

21: **procedure** $\mathrm{SETRATE}(t, \boldsymbol{x}_t, m)$
22:     **if** STOCHASTIC **then**
23:         $\eta_t = \sqrt{\frac{\ln m}{t}}$
24:     **end if**
25:     **if** ADVERSARIAL **then**
26:         Set $\rho_t$ as in adversarial setting section
27:         $\eta_t = \sqrt{\frac{1}{\sqrt{t}} + \frac{\rho_t}{c^2 \ln c}} \cdot \sqrt{\frac{\ln m}{T}}$
28:     **end if**
29:     **return** $\eta_t$
30: **end procedure**

29: **procedure** $\mathrm{RECOMMEND}(\boldsymbol{x}_t, \boldsymbol{q}_t)$
30:     **if** STOCHASTIC **then**
31:         $\boldsymbol{w}_t = \sum_{i \in |\Pi^*|} q_{t,i} \pi_i(\boldsymbol{x}_t)$
32:         $j_t \leftarrow \mathrm{maxind}(\boldsymbol{w}_t)$
33:     **end if**
34:     **if** ADVERSARIAL **then**
35:         $i_t \sim \boldsymbol{q}_t$
36:         $j_t \sim \pi_{i_t}(\boldsymbol{x}_t)$
37:     **end if**
38:     **return** $j_t$
39: **end procedure**

Figure 1: The Contextual Active Model Selection (CAMS) algorithm

**Contextual model selection.** Our key insight underlying the contextual model selection strategy extends from the *online learning with expert advice* framework [13, 27]. We start by appending the constant policies that always pick single pre-trained *models* to form the extended policy set $\Pi^*$ (Line 3, in Fig. 1). This allows CAMS to be at least as competitive as the best model. Then, at each round, CAMS maintains a probability distribution over the (extended) policy set $\Pi^*$, and updates those according to the observed loss for each policy. We use $\boldsymbol{q}_t := (q_{t,i})_{i \in |\Pi^*|}$ to denote the probability distribution over $\Pi^*$ at $t$. Specifically, the probability $q_{t,i}$ is computed based on the exponentially weighted cumulative loss, i.e. $q_{t,i} \propto \exp\left(-\eta_t \tilde{L}_{t-1,i}\right)$ where $\tilde{L}_{t,i} := \sum_{\tau=1}^{t} \tilde{\ell}_{\tau,i}$ denotes the cumulative loss of policy $i$.

---

[2]Assume ties are broken randomly.

For adversarial data streams, it is natural for both the online learner and the model selection policies to randomize their actions to avoid linear regret [33]. Following this insight, CAMS randomly samples a policy $i_t \sim \boldsymbol{q}_t$, and—based on the current context $\boldsymbol{x}_t$—samples a classifier $j_t \sim \pi_{i_t}(\boldsymbol{x}_t)$ to recommend at round $t$.

Under the stochastic setting, CAMS adopts a weighted majority strategy [45] when selecting models. The vector of each model's weighted votes from the policies, $\boldsymbol{w}_t = \sum_{i \in |\Pi^*|} q_{t,i} \pi_i(\boldsymbol{x}_t)$, is interpreted as a distribution induced by the weighted policy. The model $j_t = \text{maxind}(\boldsymbol{w}_t)$ which receives the highest probability becomes the recommended model at round $t$. This deterministic model selection strategy is commonly used in stochastic online optimization [33]. An alternative strategy is to take a randomized approach as in the adversarial setting, or take a Follow-the-Leader approach [42] and go with the most probable model recommended by the most probable policy (i.e. use $\boldsymbol{w}_t = \pi_{\text{maxind}(\boldsymbol{q}_t)}(\boldsymbol{x}_t)$).As shown in experimental results section and further discussed in Appendix (outperformance over the best policy/expert section), CAMS outperforms these policies in a wide range of practical applications. The model selection steps are detailed in Line 5-9 in Fig. 1.

**Active queries.** Under a limited budget, we intend to query the labels of those instances that exhibit significant disagreement among the pre-trained models $\mathcal{F}$. To achieve this goal, we design an adaptive query strategy with query probability $z_t$. Concretely, given context $\boldsymbol{x}_t$, model predictions $\hat{\boldsymbol{y}}_t$ and model distribution $\boldsymbol{w}_t$, we denote by $\bar{\ell}_t^y := \langle \boldsymbol{w}_t, \mathbb{I}\{\hat{\boldsymbol{y}}_t \neq y\}\rangle$ as the expected loss if the true label is $y$. We characterize the model disagreement as

$$\mathfrak{E}(\hat{\boldsymbol{y}}_t, \boldsymbol{w}_t) := \frac{1}{c} \sum_{y \in \mathcal{Y}, \bar{\ell}_t^y \in (0,1)} \bar{\ell}_t^y \log_c \frac{1}{\bar{\ell}_t^y}. \tag{3}$$

Intuitively, when $\bar{\ell}_t^y$ is close to 0 or 1, there is little disagreement among the models in labeling $\boldsymbol{x}_t$ as $y$, otherwise there is significant disagreement. We capture this insight with function $h(x) = -x \log x$. Since the label $y_t$ is unknown upfront when receiving $\boldsymbol{x}_t$, we iterate through all the possible labels $y \in \mathcal{Y}$ and take the average value as in Eq. (3). Note that $\mathfrak{E}$ takes a similar algebraic form to the entropy function, although it does not inherit the information-theoretic interpretation.

With the model disagreement term defined above, we consider an adaptive query probability[3]

$$z_t = \max\left\{\delta_0^t, \mathfrak{E}(\hat{\boldsymbol{y}}_t, \boldsymbol{w}_t)\right\}, \tag{4}$$

where $\delta_0^t = \frac{1}{\sqrt{t}} \in (0, 1]$ is an adaptive lower bound on the query probability to encourage exploration at an early stage. The query strategy is summarized in Line 10-14 in Fig. 1.

**Model updates.** Now define $U_t \sim \text{Ber}(z_t)$ as a binary query indicator that is sampled from a Bernoulli distribution parametrized by $z_t$. Upon querying the label $y_t$, one can calculate the loss for each model $f_j \in \mathcal{F}$ as $\ell_{t,j} = \mathbb{I}\{\hat{y}_{t,j} \neq y_t\}$. Since CAMS does not query all the i.i.d. examples, we introduce an unbiased loss estimator for the models, defined as $\hat{\ell}_{t,j} = \frac{\ell_{t,j}}{z_t} U_t$. The unbiased loss of policy $\pi_i \in \Pi^*$ can then be computed as $\tilde{\ell}_{t,i} = \langle \pi_i(\boldsymbol{x}_t), \hat{\ell}_{t,j}\rangle$. In the end, CAMS computes the (unbiased) cumulative loss of policy $\pi_i$ as $\tilde{L}_{T,i} = \sum_{t=1}^T \tilde{\ell}_{t,i}$, which is used to update the policy probability distribution in next round. Pseudocode for the model update steps is summarized in Line 15-21 in Fig. 1.

*Remark.* CAMS runs efficiently with time complexity $O(nk)$ per round and space complexity $O((n+k) \cdot k)$. At each round, each model selection policy specifies a probability distribution over the models for the given context. When these distributions correspond to constant Dirac delta distributions (regardless of the context), the problem reduces to the context-free problem investigated by Karimi et al. [39].

## 5 Theoretical Analysis

We now present theoretical bounds on the regret (defined in Eq. (1) and Eq. (2), respectively) and the query complexity of CAMS for both the stochastic and the adversarial settings.

---

[3]For convenience of discussion, we assume that those rounds where all policies in $\Pi^*$ select the same models or all models $\mathcal{F}$ make the same predictions are removed as a precondition.

## 5.1 Stochastic setting

Under the stochastic setting, the cumulative loss of CAMS over T rounds—as specified by the RECOMMEND procedure—is $L_T^{\text{CAMS}} = \sum_{t=1}^{T} \hat{\ell}_{t,\text{maxind}(\boldsymbol{w}_t)}$ where recall $\boldsymbol{w}_t = \sum_{i \in |\Pi^*|} q_{t,i} \pi_i(\boldsymbol{x}_t)$ is the probability distribution over $\mathcal{F}$ induced by the weighted policy.

Let $i^* = \arg\min_{i \in [|\Pi^*|]} \mu_i$ be the index of the best policy ($\mu_i$ denotes the expected loss of policy $i$, as defined in problem statement section. The cumulative expected loss of policy $i^*$ is $T\mu_{i^*}$; therefore the expected pseudo-regret (Eq. (1)) is $\overline{\mathcal{R}}_T (\text{CAMS}) = \mathbb{E}\left[\sum_{t=1}^{T} \hat{\ell}_{t,\text{maxind}(\boldsymbol{w}_t)}\right] - T\mu_{i^*}$.

Define $\Delta := \min_{i \neq i^*} (\mu_i - \mu_{i^*})$ as the minimal sub-optimality gap[4] in terms of the expected loss against the best policy $i^*$. Furthermore, let $\boldsymbol{w}_{i^*}^t := \pi_{i^*}(\boldsymbol{x}_t)$ be probability distribution over $\mathcal{F}$ induced by policy $i^*$ at round $t$. We define $\gamma := \min_{\boldsymbol{x}_t}\{\max_{w_j \in \boldsymbol{w}_{i^*}^t} w_j - \max_{w_j \in \boldsymbol{w}_{i^*}^t, j \neq \text{maxind}(\boldsymbol{w}_{i^*}^t)} w_j\}$ (5) as the minimal probability gap between the most probable model and the rest (assuming no ties) induced by the best policy $i^*$. We further define $b = p_{\min} \log_c (1/p_{\min})$, where $p_{\min} = \min_{s,i} \pi(\boldsymbol{x}_s)$ denotes the minimal model selection probability by any policy[5]. As our first main theoretical result, we show that, without exhaustively querying the labels of the stochastic stream, CAMS achieves constant expected regret.

**Theorem 1.** *(Regret) In the stochastic environment, with probability at least $1 - \delta$, CAMS achieves constant expected pseudo regret* $\overline{\mathcal{R}}_T (\text{CAMS}) = \left(\frac{\ln \frac{|\Pi^*|-1}{\gamma} + \sqrt{\ln |\Pi^*| \cdot 2b^2 \ln \frac{2}{\delta}}}{\sqrt{\ln |\Pi^*|} \Delta}\right)^2$.

Note that in the stochastic setting, a lower bound of $\Omega\left((\log \Pi^*)/\Delta\right)$ was shown in Mourtada and Gaïffas [50] for online learning problems with expert advice under the full information setting (i.e. assuming labels are given for all data points in the stochastic stream). To establish the proof of Theorem 1, we consider a novel procedure to connect the weighted policy by CAMS to the best policy $\pi_{i^*}$. Conceptually, we would like to show that, after a *constant* number of rounds $\tau_{\text{const}}$, with high probability, the model selected by CAMS (Line 32) will be the same as the one selected by the best policy $i^*$. In that way, the expected pseudo regret will be dominated by the maximal cumulative loss up to $\tau_{\text{const}}$. Toward this goal, we first bound the weight of the best policy $w_{t,i^*}$ as a function of $t$, by choosing a proper learning rate $\eta_t$ (CAMS, Line 23). Then, we identify a constant threshold $\tau_{\text{const}}$, beyond which CAMS exhibits the same behavior as $\pi_{i^*}$ with high probability. Finally, we obtain the regret bound by inspecting the regret at the two stages separately. The formal statement of Theorem 1 and the detailed proof are deferred to App. E.1.

Next, we provide an upper bound on the query complexity in the stochastic setting.

**Theorem 2.** *(Query Complexity). For c-class classification problems, with probability at least $1 - \delta$, the expected number of queries made by* CAMS *over T rounds is upper bounded by* $\left(\left(\frac{\ln \frac{|\Pi^*|-1}{\gamma} + \sqrt{\ln |\Pi^*| \cdot 2b^2 \ln \frac{2}{\delta}}}{\sqrt{\ln |\Pi^*|} \Delta}\right)^2 + T\mu_{i^*}\right) \frac{\ln T}{c \ln c}$.

Theorem 2 is built upon Theorem 1, where the the key idea behind the proof is to relate the number of updates to the regret. When $T\mu_{i^*}, \tilde{L}_{T,*}$ are regarded as constants (given by an oracle), the query-complexity bound is then sub-linear *w.r.t.* $T$. Note that the number of class labels $c$ affects the quality of the query complexity bound. The intuition behind this result is, with larger number of classes, *each query may carry more information upon observation*. For instance, in an extreme case where only one expert always recommends the best model and others gives random recommendations of models (and predicts random labels), having more classes lowers the chance of a model making the correct guess, and therefore helps to "filter out" those suboptimal experts in fewer rounds—hence being more query efficient. We defer the proof of Theorem 2 to App. E.2.

## 5.2 Adversarial setting

Now we consider the adversarial setting. Let $\tilde{L}_{T,*} := \min_{i \in [|\Pi^*|]} \sum_{t=1}^{T} \tilde{\ell}_{t,i}$ be the cumulative loss of the best policy. The expected regret (Eq. (2)) for CAMS equals to $\mathcal{R}_T (\text{CAMS}) =$

---

[4]w.l.o.g. assume there is a single best policy, and thus $\Delta > 0$.

[5]We assume $p_{\min} > 0$ per the policy regularization criterion in Appendix C.3. (cf. Algorithm 1 on "Regularized policy $\bar{\pi}(\boldsymbol{x}_t)$)".

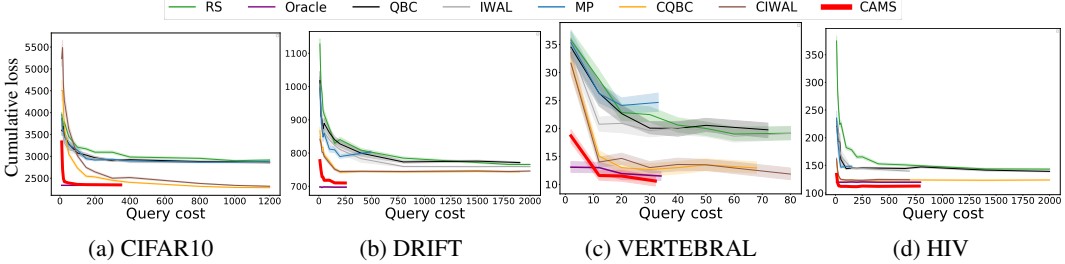

Figure 2: **Main results.** Comparison of CAMS with 7 baselines across 4 diverse benchmarks in terms of cost effectiveness. We plot the cumulative loss as we increase the query cost for a fixed number of rounds $T$ and maximal query cost $B$ (from left to right: $T = 10000, 3000, 80, 4000$, and $B = 1200, 2000, 80, 2000$). CAMS outperforms all baselines. **Algorithms**: 4 contextual {Oracle, CQBC, CIWAL, CAMS} and 4 non-contextual baselines {RS, QBC, IWAL, MP} are included (see Section ). 90% confident interval are indicated in shades.

$\mathbb{E}\big[\sum_{t=1}^{T}\langle \boldsymbol{q}_t, \tilde{\boldsymbol{\ell}}_t\rangle\big] - \tilde{L}_{T,*}$. We show that under the adversarial setting, CAMS achieves sub-linear regret in $T$ without accessing all labels.

**Theorem 3.** *(Regret) Let $c$ be the number of classes and $\rho_t$ be specified as Line 26-27 in the* SETRATE *procedure. Under the adversarial setting, the expected regret of* CAMS *is bounded by* $2c\sqrt{\ln c/\max\{\rho_T, \sqrt{1/T}\}} \cdot \sqrt{T\log|\Pi^*|}.$

The proof is provided in App. F.1. Assuming $\rho_t$ to be a constant, our regret upper bound in Theorem 3 matches (up to constants) the lower bound of $\Omega\left(\sqrt{T\ln|\Pi^*|}\right)$ for online learning problems with expert advice under the full information setting [15, 63] (i.e. assuming labels are given for all data points). Hereby, the decaying learning rate $\eta_t$ as specified in Line 27 is based on two parameters, where $1/\sqrt{t}$ corresponds to the lower bound $\delta_0^t$ on the query probability, and $\rho_t \triangleq 1 - \max_{\tau \in [t-1]}\langle \boldsymbol{w}_\tau, \mathbb{I}\{\hat{\boldsymbol{y}}_\tau = y_\tau\}\rangle$ (6) is a (data-dependent) term that is chosen to reduce the impact of the randomized query strategy on the regret bound (especially when $t$ is large). Intuitively, $\rho_t$ relates to the skewness of the policy where the max term corresponds to the maximal probability of most probable mispredicted label over $t$ rounds. Note that in theory $\rho_t$ can be small (e.g. CAMS may choose a constant policy $\pi_i^{\text{const}} \in \Pi^*$ that mispredict the label for $\boldsymbol{x}_t$, which leads to $\rho_t = 0$); in such cases, our result still translates to a sublinear regret bound of $O(c\sqrt{\log c} \cdot T^{\frac{3}{4}}\sqrt{\log|\Pi^*|})$. Furthermore, in practice, we consider to "regularize" the policies (App. D.4) to ensure that probability a policy selecting any model is bounded away from 0.

Finally, the following theorem (proof in App. F.2) establishes a query complexity bound of CAMS.

**Theorem 4.** *(Query Complexity). Under the adversarial setting, the expected query complexity over $T$ rounds is* $O\left(\ln T\left(\sqrt{\dfrac{T\log|\Pi^*|}{\max\{\rho_T, \sqrt{1/T}\}}} + \tilde{L}_{T,*}\right)\right).$

## 6 Experiments

**Datasets.** We evaluate our approach using five datasets: (1) CIFAR10 [41] contains 60,000 images from 10 different balanced classes. (2) DRIFT [73] is a tabular dataset with 128-dimensional features, based on 13,910 chemical sensor measurements of 6 types of gases at various concentration levels. (3) VERTEBRAL [5] is a biomedical tabular dataset which classifies 310 patients into three classes (Normal, Spondylolisthesis, Disk Hernia) based on 6 attributes. (4) HIV [74] contains over 40,000 compounds annotated with molecular graph features and binary labels (active, inactive) indicating their ability to inhibit HIV replication. (5) CovType [24] has 580K samples and contains details including slope, aspect, elevation, measurements of area, and type of forest cover.

**Policy sets.** We construct the policy sets $\Pi$ for each dataset following a procedure similar to Meta-selector [48]. In this approach, a set of recommender algorithms is considered, and Meta-selector assigns varying ratings to these algorithms based on the specific user. Concretely, we first construct a set of models trained on different subsamples from each dataset. We then construct a set of policies, which include *malicious*, *normal*, *random*, and *biased* policy types for each dataset

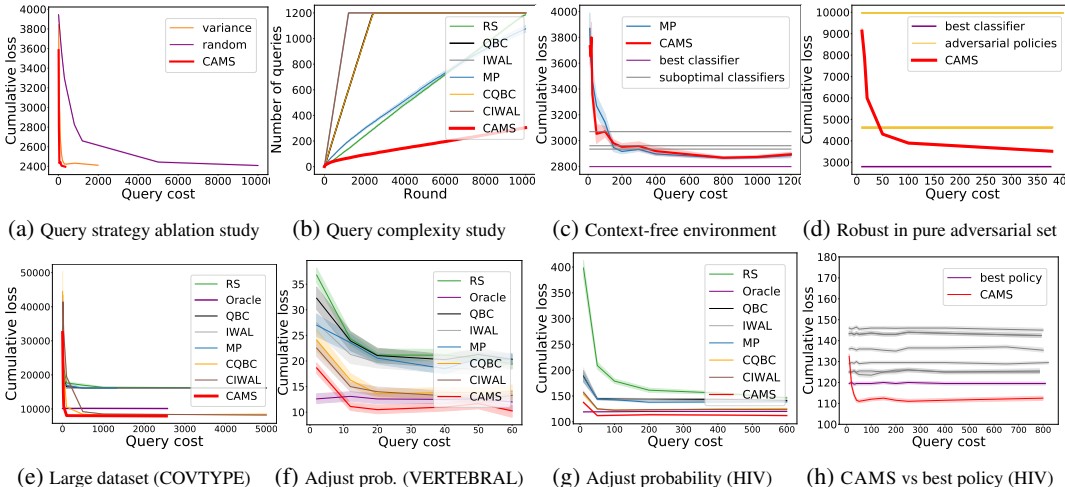

(a) Query strategy ablation study    (b) Query complexity study    (c) Context-free environment    (d) Robust in pure adversarial set

(e) Large dataset (COVTYPE)    (f) Adjust prob. (VERTEBRAL)    (g) Adjust probability (HIV)    (h) CAMS vs best policy (HIV)

Figure 3: **Ablation studies.** (a) Comparing three query strategies {CAMS, variance-based, random} under same model selection policy. (b) Comparing the increasing rate of CAMS' query cost over other baselines. (c) Comparing CAMS with MP in context-free environment. (d) Evaluating the performance of CAMS under a pure adversarial setting. (e) Large dataset. (f,g) Adjustable query probability. (h) CAMS outperforms the best single policy. The ablation study (a)-(d) is conducted on CIFAR10. For additional results on other benchmarks, please refer to the supplemental material.

based on different models and features. Details on the classifiers and policies are provided in the supplemental materials. The *malicious* policy provides contrary advice; the *random* policy provides random advice; the *biased* policy provides biased advice by training on a biased distribution for classifying specific classes. The *normal* policy gives reasonable advice, being trained under a standard process on the training set. We represent the output of the $i_{th}$ policy as $\pi_i(\boldsymbol{x}_t)$, indicating the rewards distribution of all the base classifiers on $\boldsymbol{x}_t$. In total, we create 80, 10, 6, 4 classifiers and 85, 11, 17, 20 policies for CIFAR10, DRIFT, VERTEBRAL, and HIV, respectively.

**Baselines.** We evaluate CAMS against both *contextual* and *non-contextual* active model selection baselines. We consider four *non-contextual* baselines: (1) Random Query Strategy (RS) which queries the instance label with a fixed probability $\frac{b}{T}$; (2) Model Picker (MP) [39] that employs variance-based active sampling with a coin-flip query probability $\max\{v(\hat{\boldsymbol{y}}_t, \boldsymbol{w}_t), \eta_t\}$, where the variance term is defined as $v(\hat{\boldsymbol{y}}_t, \boldsymbol{w}_t) = \max_{y\in\mathcal{Y}} \bar{\ell}_t^y (1 - \bar{\ell}_t^y)$; (3) Query by Committee (QBC) implementing committee-based sampling [22]; and (4) Importance Weighted Active Learning (IWAL) [8] that calculates query probability based on labeling disagreements of surviving classifiers. Since no *contextual* baselines exist yet, we propose contextual versions of QBC and IWAL as (5) CQBC and (6) CIWAL. Both extensions maintain their respective original query strategies but incorporate the context into the cumulative rewards. For *model selection*, CAMS, MP, CQBC, and CIWAL recommend the classifier with the highest probability. The other baselines use Follow-the-Leader (FTL), recommending the model with the minimum cumulative loss for past queried instances. Finally, we add (7) Oracle to represent the best single policy with the minimum cumulative loss, with the same query strategy as CAMS.

### 6.1 Main results

Fig. 2 visualizes the *cost effectiveness* of CAMS and the baselines. Here, we define *cost effectiveness* as the measure of how quickly the cumulative loss decreases in response to an increase in query cost. Fig. 2 demonstrates that CAMS outperforms all the comparison methods across all benchmarks. Remarkably, it outperforms even the oracle on the VERTEBRAL (Fig. 2c) and HIV (Fig. 2d) benchmarks with fewer than 10 and 20 queries, respectively. In the case of the VERTEBRAL benchmark, CAMS outperforms the best baseline in query cost by a margin of $20\%$, despite the fact that 11 out of the 17 experts provided malicious or random advice. This level of performance is attained by utilizing an active query strategy to retrieve highly informative data, thereby maximizing the differentiation between models and policies within the constraints of a limited budget. Additionally, the model selection strategy allows for effectively combining the expertise among the experts.

## 6.2 Ablation studies

**Effectiveness of active querying.** In Fig. 3a and Fig. 3b, we perform ablation studies to demonstrate the effectiveness of our active query strategy. We fix the model recommendation strategy as the one used by CAMS, and compare three query strategies: (1) CAMS, (2) the state-of-the-art variance-based query strategy from Model Picker [39] (referred to as "variance"), and (3) a random query strategy. Figure 3a demonstrates that CAMS has the fastest convergence rate in terms of cumulative loss on CIFAR10, implying effective use of queried labels. Furthermore, CAMS not only achieves the minimum cumulative loss but also incurs significantly lower query costs, with reductions of 71% and 95% compared to the variance and random strategies respectively as showed in Fig. 3b. This suggests that CAMS selectively queries data to optimize policy improvement, whereas the other strategies may query unnecessary labels, including potentially noisy or uninformative ones, which impede policy improvement and convergence.

**Robustness.** In Fig. 2, 3c, 3d,3e, 3f, and 3g, CAMS exhibits robustness in a variety of environmental settings. Firstly, As shown in Fig. 2, CAMS outshines other methods in a contextual environment, whereas in Fig. 3c, a non-contextual (no experts) environment, it achieves comparable performance to the state-of-the-art Model Picker in identifying the best classifier. Secondly, CAMS is robust in both stochastic and adversarial environments. As demonstrated in Fig. 2, CAMS surpasses other methods in a stochastic environment. Additionally, as illustrated in Fig. 3d, in a worst-case adversarial environment, CAMS effectively recovers from adversarial actions and approaches the performance of the best classifier (see App. G.5). We further observe that CAMS demonstrates robustness to varying scales of data, where the online stream sizes range from 80 to 10K (Fig. 2) to 100K (Fig. 3e, where we randomly sample 100K samples from the CovType dataset [24]).

In Fig. 2, we assume that the stream length $T$ is hidden and not used as input to CAMS. Under the stochastic setting, however, knowing $T$ can provide additional information that one can leverage to optimize the query probability, thereby giving an advantage to some of the baseline algorithms (e.g. random). As an ablation study, in Fig. 3f and Fig. 3g, we assume the stochastic setting where the total length $T$ of the online stream is given. Given the stream length $T$ and query budget $b$, we may optimize each algorithm by scaling their query probabilities, so that each algorithm allocates its query budget to the top $b$ informative labels in the entire online stream based on its own query criterion. CAMS still ourperform the baselines under the setting.

**Improvement over the best classifier and policy.** Fig. 3h demonstrates that when provided with good policies, CAMS formulates a stronger policy which incurs no regret. CAMS has the potential to outperform an oracle, especially in rounds where the oracle does not make the optimal recommendation. For instance, in the stochastic version of CAMS (as shown in lines 22-23 and 30-32 of Fig. 1), CAMS recommends a model using a weighted majority vote among all policies, enabling the formation of a new policy in each round by amalgamating the strengths of each sub-optimal policy. This adaptive strategy can potentially outperform any single policy. Moreover, in most real-world scenarios and conducted experiments (as depicted in App. G.6), data streams may not be strictly stochastic, and therefore no single policy consistently performs the best. In such cases, CAMS's weighted policy may find an enhanced combination of "advices", leading to improved performance.

## 7 Conclusion

We introduced CAMS, an online contextual active model selection framework based on a novel model selection and active query strategy. The algorithm was motivated by many real-world use cases that need to make decision by taking both contextual information and the cost into consideration. We have demonstrated CAMS's compelling performance of using the minimum query cost to learn the optimal contextual model selection policy on several diverse online model selection tasks. In addition to the promising empirical performance, we also provided rigorous theoretical guarantees on the regret and query complexity for both stochastic and adversarial settings. We hope our work can inspire future works to handle more complex real-world model selection tasks (e.g. beyond classification or non-uniform loss functions, etc. where our analysis does not readily apply).

**Acknowledgements**

This work is supported in part by the RadBio-AI project (DE-AC02-06CH11357), U.S. Department of Energy Office of Science, Office of Biological and Environment Research, the IMPROVE project under contract (75N91019F00134, 75N91019D00024, 89233218CNA000001, DE-AC02-06-CH11357, DE-AC52-07NA27344, DE-AC05-00OR22725), the Laboratory Directed Research and Development (LDRD) funding from Argonne National Laboratory provided by the Director, Office of Science, of the U.S. Department of Energy under Contract No. DE-AC02-06CH11357, the Exascale Computing Project (17-SC-20-SC), a collaborative effort of the U.S. Department of Energy Office of Science and the National Nuclear Security Administration, the University of Chicago Joint Task Force Initiative, the AI-Assisted Hybrid Renewable Energy, Nutrient, and Water Recovery project (DOE DE-EE0009505), and the National Science Foundation under Grant No. IIS 2313131 and IIS 2332475.

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

# A  Impact Statements

This paper introduces a novel framework for adaptive model selection in label-efficient learning. By integrating robust online learning with active query strategies, our algorithm effectively adapts to varying data contexts and minimizes labeling efforts, crucial in domains requiring swift and accurate decisions, such as disease identification and financial predictions. Ethically, the framework's design promotes efficient and context-aware model selection, reducing potential biases associated with context-ignorant model selections. No major ethical concerns are anticipated, given the algorithm's generality and focus on solving practical problems.

# B  Table of Notations Defined in the Main Paper

| notation | meaning |
|---|---|
| **Problem Statement** | |
| $\mathcal{X}$ | input domain |
| $\boldsymbol{x}$ | input instance, $\boldsymbol{x} \in \mathcal{X}$ |
| $t, T$ | index of a round, total number of rounds |
| $i, j$ | index of policies, models/classifiers |
| $n$ | number of policies |
| $k$ | number of classifiers |
| $\mathcal{Y}$ | $\{0, \ldots, c-1\}$, set of $c$ possible class labels for each input instance |
| $c$ | number of class labels, $|\mathcal{Y}|$ |
| $\Delta^{k-1}$ | $k$-dimensional probability simplex $\{\boldsymbol{w} \in \mathbb{R}^k : |\boldsymbol{w}| = 1, \boldsymbol{w} \geq 0\}$ |
| $f$ | single pre-trained classifier (model) |
| $\mathcal{F}$ | $\{f_1, \ldots, f_k\}$, set of $k$ pre-trained classifiers over $\mathcal{X} \times \mathcal{Y}$ |
| $\pi, \pi(\boldsymbol{x})$ | model selection policy (expert) $\pi : \mathcal{X} \to \Delta^{k-1}$, probability of selecting each classifier under input $\boldsymbol{x}$ |
| $\pi^{\text{const}}$ | $\pi_j^{\text{const}}(\cdot) := \boldsymbol{e}_j, \boldsymbol{e}_j \in \Delta^{k-1}$ denotes the canonical basis vector with $e_j = 1$ |
| $\Pi$ | collection of model selection policies |
| $\Pi^*$ | $\Pi \cup \{\pi_1^{\text{const}}, \ldots, \pi_k^{\text{const}}\}$, extended policy set including constant policies that always suggest a fixed model |
| $|\Pi|, |\Pi^*|$ | $n, |\Pi^*| \leq (n+k)$ |
| $\pi^*$ | $\pi^* \in \Pi^*$, best policy |
| $\hat{y}_{t,j}$ | $f_j(\boldsymbol{x}_t)$, predicted label for $j_{\text{th}}$ pre-trained classifier at round $t$ |
| $y_t$ | true label of $\boldsymbol{x}_t$ |
| $\hat{\boldsymbol{y}}_t$ | $[\hat{y}_{t,1}, \ldots, \hat{y}_{t,k}]^\top$, predicted labels by all $k$ models at round t |
| $\ell_{t,j}$ | $\mathbb{I}_{\{\hat{y}_{t,j} \neq y_t\}}$, 0-1 loss for model $j \in [k]$ at round $t$ |
| $\boldsymbol{\ell}_t$ | $\mathbb{I}_{\{\hat{\boldsymbol{y}}_t \neq y_t\}}$, full loss vector upon observing $y_t$ |
| $\mathcal{A}$ | the learner |
| $L_T^{\mathcal{A}}$ | $\sum_{t=1}^T \ell_{t,j_t}$, cumulative loss over $T$ rounds for a learning algorithm $\mathcal{A}$ |
| $\tilde{\ell}_{t,i}$ | $\langle \pi_i(\boldsymbol{x}_t), \boldsymbol{\ell}_t \rangle$, expected loss if the learner commits to policy $\pi_i$ and take random selection at round $t$ |
| $\text{maxind}(\boldsymbol{w})$ | $\arg\max_{j, w_j \in \boldsymbol{w}} w_j$, index of maximal value entry of $\boldsymbol{w}$ |
| $\mu$ | $\frac{1}{T} \sum_{t=1}^T \mathbb{E}_{\boldsymbol{x}_t, y_t} \left[ \hat{\ell}_{t, \text{maxind}(\pi_i(\boldsymbol{x}_t))} \right]$ |
| $\mathcal{R}_T(\mathcal{A}), \overline{\mathcal{R}}_T(\mathcal{A})$ | expected regret in adversarial setting, pseudo-regret for stochastic setting |
| $\mathbb{E}_t[\cdot]$ | $\mathbb{E}[\cdot|\mathcal{F}_t], \mathcal{F}_t = \sigma\left(\boldsymbol{E}^{(1)}, \hat{\boldsymbol{y}}_1, \ldots, \hat{\boldsymbol{y}}_{t-1}, \boldsymbol{E}^{(t)}\right)$ |
| **Algorithm** | |
| $\boldsymbol{q}_t$ | $(q_{t,i})_{i \in |\Pi^*|}$, probability distribution over $\Pi^*$ at $t$ |
| $\tilde{L}_{t,i}$ | $\sum_{\tau=1}^t \tilde{\ell}_{\tau,i}$, cumulative loss of policy $i$ |
| $\boldsymbol{w}_t$ | $\sum_{i \in |\Pi^*|} q_{t,i} \pi_i(\boldsymbol{x}_t)$, distribution induced by the weighted policy |
| $\bar{\ell}_t^y$ | $\langle \boldsymbol{w}_t, \mathbb{I}\{\hat{\boldsymbol{y}}_t \neq y\} \rangle$, expected loss if the true label is $y$ |
| $\mathfrak{E}(\hat{\boldsymbol{y}}_t, \boldsymbol{w}_t)$ | model disagreement function |
| $h(x)$ | $-x \log x$ |
| $\delta_0^t$ | $\frac{1}{\sqrt{t}}$, lower bound of query probability |
| $z_t$ | $\max\{\delta_0^t, \mathfrak{E}(\hat{\boldsymbol{y}}_t, \boldsymbol{w}_t)\}$, adaptive query probability |
| $\hat{\ell}_{t,j}$ | $\frac{\ell_{t,j}}{z_t} U_t$ |
| $U$ | query indicator |
| $\eta_t$ | adaptive learning rate |
| $\rho_t$ | $1 - \max_{\tau \in [t-1]} \langle \boldsymbol{w}_\tau, \mathbb{I}\{\hat{\boldsymbol{y}}_\tau = y_\tau\} \rangle$ |
| $b$ | query budget |
| $\hat{\boldsymbol{\ell}}, \left(\hat{\ell}_{t,i}\right)_{i \in [k]}$ | unbiased estimate of classifier loss vector |
| $\widetilde{\boldsymbol{\ell}}, \left(\widetilde{\ell}_{t,i}\right)_{i \in [n]}$ | unbiased estimate of policy loss vector |
| $\widehat{\boldsymbol{L}}$ | unbiased cumulative loss of classifiers, policies |
| **Analysis** | |
| $p_{t,y}$ | $\sum_{j \in [k]} \mathbb{I}\{\hat{y}_{t,j} = y\} w_j$, the total probability of classifiers predicts label $y$ at round $t$ |
| $\Delta$ | $\min_{i \neq i^*} \Delta_i = \min_{i \neq i^*}(\mu_i - \mu_{i^*})$, sub-optimality gap |
| $\gamma$ | $\min_{\boldsymbol{x}_t} \left\{ \max_{w_j \in \boldsymbol{w}_{i^*}^t} w_j - \max_{w_j \in \boldsymbol{w}_{i^*}^t, j \neq \text{maxind}(\boldsymbol{w}_{i^*}^t)} w_j \right\}$, sub-optimality model probability gap of $\pi_{i^*}$ |
| $\Delta_i$ | $\mathbb{E}[\widetilde{\ell}_{\cdot,i} - \widetilde{\ell}_{\cdot,i^*}]$, sub-optimality gap or immediate regret of policy $i$ |
| $L_{T,*}$ | the cumulative loss of oracle at round T |

Table 2: Notations used in the main paper

## C  Summary of Regret and Query Complexity Bounds

We summarize the regret and query complexity bounds (if applicable) of related algorithms in Table 3.

| Algorithm | Regret | Query Complexity |
|---|---|---|
| Exp3 [42] | $2\sqrt{Tk\log k}$ | – |
| Exp3.p [12] | $5.15\sqrt{nT\log\frac{n}{\delta}}$ | – |
| Exp4 [42] | $\sqrt{2Tk\log n}$ | – |
| Exp4.p [9] | $6\sqrt{kT\ln\frac{n}{\delta}}$ | – |
| Model Picker$_{\text{stochastic}}$ [39] | $62\max_i \Delta_i k/\left(\lambda^2\log k\right)$ $\lambda = \min_{j\in[k]\setminus\{i^*\}}\Delta_j^2/\theta_j$ | $\sqrt{2T\log k}(1+4\frac{c}{\Delta})$ |
| Model Picker$_{\text{adversarial}}$ [39] | $2\sqrt{2T\log k}$ | $5\sqrt{T\log k}+2L_{T,*}$ |
| **CAMS**$_{\text{STOCHASTIC}}$ | $\left(\dfrac{\ln\frac{\|\Pi^*\|-1}{\gamma}+\sqrt{\ln\|\Pi^*\|\cdot 2b^2\ln\frac{2}{\delta}}}{\sqrt{\ln\|\Pi^*\|}\Delta}\right)^2$ | $\left(\left(\dfrac{\ln\frac{\|\Pi^*\|-1}{\gamma}+\sqrt{\ln\|\Pi^*\|\cdot 2b^2\ln\frac{2}{\delta}}}{\sqrt{\ln\|\Pi^*\|}\Delta}\right)^2+T\mu_{i^*}\right)\dfrac{\ln T}{c\ln c}$ |
| **CAMS**$_{\text{ADVERSARIAL}}$ | $2c\sqrt{\ln c/\max\{\rho_T,\sqrt{1/T}\}}\cdot\sqrt{T\log\|\Pi^*\|}$ | $O\left(\left(\sqrt{\dfrac{T\log\|\Pi^*\|}{\max\{\rho_T,\sqrt{1/T}\}}}+\tilde{L}_{T,*}\right)(\ln T)\right)$ |

Table 3: Regret and query complexity bounds. For the notations in this table: $i^*$ is the model with the highest expected accuracy; $\theta_j = \mathbb{P}\left[\ell_{.,j}\neq\ell_{.,i^*}\right]$ is the probability that exactly one of $j$ and $i^*$ correctly classifies a sample; $\gamma$ and $\rho_T$ are defined in Eq. (5) and (6), respectively. $b = p_{\min}\log_c\left(1/p_{\min}\right)$, where $p_{\min} = \min_{s,i}\pi(\mathbf{x}_s)$ denotes the minimal model selection probability by any policy.

*Remark* 5. When $T\mu_{i^*}, \tilde{L}_{T,*}$ are regarded as constants (given by an oracle), the query-complexity bound is then sub-linear *w.r.t.* $T$.

*Remark* 6. Note that the number of class labels $c$ affects the quality of the query complexity bound. The intuition behind this result is, with larger number of classes, *each query may carry more information upon observation*. For instance, in an extreme case where only one expert always recommends the best model and others gives random recommendations of models (and predicts random labels), having more classes lowers the chance of a model making the correct guess, and therefore helps to "filter out" those suboptimal experts in fewer rounds—hence being more query efficient.

*Remark* 7. To prove the practical feasibility of CAMS, we have analyzed its time and space complexity. Our analysis shows that CAMS has a time complexity of $O\left(Tnk\right)$ in total or $O(nk)$ per round (due to the RECOMMEND procedure under the stochastic setting), and a space complexity of $O\left((n+k)\cdot k\right)$. Here, $T$ refers to the online horizon, $n$ denotes the number of policies, and $k$ denotes the number of models. Taking into account these complexities, we can confirm that CAMS is practically feasible.

## D  Supplemental Materials on Experimental Setup

### D.1  Baselines

**Model Picker (MP)**  Model Picker [39] is a context-free online active model selection method inspired by EXP3. Model Picker aims to find the best classifier in hindsight while making a small number of queries. For query strategy, it uses a variance-based active learning sampling method to select the most informative label to query to differentiate a pool of models, where the variance is defined as $v\left(\hat{\mathbf{y}}_t, \mathbf{w}_t\right) = \max_{y\in\mathcal{Y}}\bar{\ell}_t^y\left(1-\bar{\ell}_t^y\right)$. The coin-flip query probability is defined as $\max\left\{v\left(\hat{\mathbf{y}}_t, \mathbf{w}_t\right), \eta_t\right\}$ when $v\left(\hat{\mathbf{y}}_t, \mathbf{w}_t\right)\neq 0$, or 0 otherwise. For model recommendation, it uses an exponential weight algorithm to recommend the model with minimal exponential cumulative loss based on the past queried labels at each round.

**Query by Committee (QBC)**  For query strategy, we have adapted the method of [22] as a disagreement-based selective sampling query strategy for online streaming data. We treat each classifier as a committee member and compute the query probability by measuring disagreement between models for each instance. The query function is coin-flip by vote entropy probability $-\frac{1}{\log\min(k,\|C\|)}\sum_c\frac{V(c,x)}{k}\log\frac{V(c,x)}{k}$, where $V(c,x)$ stands for the number of committee members assigning a class c for input context x and k is the number of committee. For the model recommendation part, we use the method of Follow-the-Leader (FTL) [42], which greedily recommends the model with the minimum cumulative loss for past queried instances.

**Importance Weighted Active Learning (IWAL)**    We have implemented [8] as the IWAL baseline. For the query strategy part, IWAL computes an adaptive rejection threshold for each instance and assigns an importance weight to each classifier in the hypothesis space $\mathcal{H}_t$. IWAL retains the classifiers in the hypothesis space according to their weighted error versus the current best classifier's weighted error at round $t$. The query probability is calculated based on labeling disagreements of surviving classifiers through function $\max_{i,j \in \mathcal{H}_t, y \in [c]} \ell_{t,i}^{(y)} - \ell_{t,j}^{(y)}$. For model recommendation, we also adopt the Follow-the-Leader (FTL) strategy.

**Random Query Strategy (RS)**    The RS method queries the label of incoming instances by the coin-flip fixed probability $\frac{b}{T}$. It also uses the FTL strategy based on queried instances for model recommendation.

**Contextual Query by Committee (CQBC)**    We have created a contextual variant of QBC termed CQBC, which has the same entropy query strategy as the original QBC. For model recommendation, we combine two model selection strategies. The first strategy calculates the cumulative reward of each classifier based on past queries and normalizes it as a probability simplex vector. We also adopt Exp4's arm recommending vector to use contextual information. Finally, we compute the element-wise product of the two vectors and normalize it to be CQBC's model recommendation vector. At each round, CQBC would recommend the top model based on the classifiers' historical performance on queried instances and the online advice matrix for streaming data.

**Contextual Importance Weighted Active Learning (CIWAL)**    We have created a variant version of importance-weighted active learning. Similar to CQBC, CIWAL adopts the query strategy from IWAL and converts the model selection strategy to be contextual. For model selection, we incorporate Exp4's arm recommendation strategy based on the side-information advice matrix and each classifier's historical performance according to queried instances. We compute the element-wise product of the two vectors as the model selection vector of CIWAL and normalize it as a weighted vector. Finally, CIWAL recommends the classifier with the highest weight.

**Oracle:**    Among all the given policies, oracle represents the best single policy that achieves the minimum cumulative loss, and it has the same query strategy as CAMS.

### D.2 Details on policies and classifiers

We constructed different expert-model configurations to reflect the cases in real-world applications[6]. This section lists the collection of policies and models used in our experiments.

**CIFAR10:**    We have constructed 80 diversified classifiers based on VGG [67], ResNet [34], DenseNet [38], GoogLeNet [69]. We have also used EfficientNet [70], MobileNets [37], RegNet [62], and ResNet to construct 85 diversified policies.

**DRIFT:**    We have constructed ten classifiers using Decision Tree [55], SVM [18], AdaBoost [28], Logistic Regression [20], KNN [19] models. We have also created 8 diversified policies with multilayer perceptron (MLP) models of different layer configurations: (128, 30, 10); (128, 60, 30, 10); (128, 120, 30, 10); (128, 240, 120, 30, 10).

**VERTEBRAL:**    We have built six classifiers using Random Forest [10], Gaussian Process [56], linear discriminant analysis [26], Naive Bayes [31] algorithms. We have constructed policies by using standard scikit-learn built-in models including Random Forest Classifier, Extra Trees Classifier [30], Decision Tree Classifier, Radius Neighbors Classifier [51], Ridge Classifier [57] and K-Nearest-Neighbor classifiers.

---

[6]To list a few other scenarios beyond the ones used in the paper: In *healthcare*, models could be the treatments, experts could be the doctors and the context could be the condition of a patient. For any patient (context), doctors (experts) will have their own advice on the treatment (model) recommendation for this patient based on their past experience. In the *finance* domain, models could be trading strategies, experts could be portfolio managers, and the context could be the stock/equity. Some trading strategies (models) might work well for the information technology sector, and some other models might work well for the energy sector, so depending on the sector of stock (context), different portfolio managers (experts) might have their own advice on different trading strategies (models) based their past trading experience.

**HIV:** We have used graph convolutional networks (GCN) [40], Graph Attention Networks (GAT) [72], AttentiveFP [75], and Random Forest to construct 4 classifiers. We have also used various feature representations of molecules such as MACCS key [25], ECFP2, ECFP4, and ECFP6 [58] molecular fingerprints to build 6 MLP-based policies, respectively.

**CovType:** We have built 6 classifiers using Random Forest, Gaussian Process, linear discriminant analysis, Naive Bayes algorithms. We have constructed 17 policies by using standard scikit-learn built-in models including Random Forest Classifier, Extra Trees Classifier, Decision Tree Classifier, Radius Neighbors Classifier, Ridge Classifier and K-Nearest-Neighbor classifiers.

### D.3 Implementation details

We build our evaluation pipeline on top of prior work [39] around the four benchmark datasets. Specifically,

- *Context $x_t$* is the raw context of the data (e.g., the 32x32 image for CIFAR10).
- *Predictions $\hat{y}_t$* contain the predicted label vector of all the classifiers' predictions according to the online context $x_t$.
- *Oracle* contains the true label $y_t$ of $x_t$.
- *Advice matrix* contains all policies' probability distribution $\lambda$ over all the classifiers on context $x_t$.

To adapt to an online setting, we sequentially draw random $T$ i.i.d. instances $x_{1:T}$ from the test pool and define it as a realization. For a fair comparison, all algorithms receive data instances in the same order within the same realization.

---

**Algorithm 1** Regularized policy $\overline{\pi}(x_t)$

---

1: **Input:** context $x_t$, Models $\mathcal{F}$, policy $\pi \in \Pi^*$
2: $\eta = \sum_{j=1}^{|\mathcal{F}|} \left( [\pi(x_t)]_j - \frac{1}{|\mathcal{F}|} \right)^2$
3: **return** $\frac{\pi_i(x_t)+\eta}{1+|\mathcal{F}|\cdot\eta}$

---

### D.4 Regularized policy

As discussed in adversarial section, we wish to ensure that the probability a policy selecting any model is bounded away from 0 so that the regret bound in Theorem 3 is non vacuous. In our experiments, we achieve this goal by applying a regularized policy $\overline{\pi}$ as shown in Algorithm 1.

### D.5 Summary of datasets and models

We summarize the attributes of datasets, the models, and the model selection policies as follows.

| dataset | classification | total instances | test set | stream size | classifier | policy |
|---------|---------------|-----------------|----------|-------------|------------|--------|
| CIFAR10 | 10 | 60000 | 10000 | 10000 | 80 | 85 |
| DRIFT | 6 | 13910 | 3060 | 3000 | 10 | 11 |
| VERTEBRAL | 3 | 310 | 127 | 80 | 6 | 17 |
| HIV | 2 | 40000 | 4113 | 4000 | 4 | 20 |
| CovType | 55 | 580000 | 100000 | 100000 | 6 | 17 |

Table 4: Attributes of benchmark datasets

### D.6 Hyperparameters

We performed our experiments on a Linux server with 80 Intel(R) Xeon(R) Gold 6148 CPU @ 2.40GHz and total 528 Gigabyte memory.

By considering the resource of server, We set 100 realizations and 3000 stream-size for DRIFT, 20 realizations and 10000 stream-size for CIFAR10, 200 realizations and 4000 stream size for HIV,

300 realization and 80 stream-size for VERTEBRAL. In each realization, we randomly selected stream-size aligned data from testing-set and make it as online streaming data which is the input of each algorithm. Thus, we got independent result for each realization.

A small realization number would increase the variance of the results due to the randomness of stream order. A large realization number would make the result be more stable but at the cost of increasing computational cost (time, memory, etc.). We chose the realization number by balancing both aspects.

# E   Proofs for the Stochastic Setting

In this section, we focus on the stochastic setting. We first prove the regret bound presented in Theorem 1 and then prove the query complexity presented in Theorem 2 for Algorithm 1.

## E.1   Proof of Theorem 1

Before providing the proof of Theorem 1, we first introduce the following lemma.

**Lemma 8.** *Fix $\tau \in (0,1)$. Let $q_{t,i^*}$ be the probability of the optimal policy $i^*$ maintained by Algorithm 1 at $t$, and let $b = p_{\min} \log_c (1/p_{\min})$, where $p_{\min} = \min_{s,i} \pi(\mathbf{x}_s)$ denotes the minimal model selection probability by any policy[7]. When $t \geq \left( \frac{\ln \frac{(|\Pi^*|-1)\tau}{1-\tau}}{\sqrt{\ln |\Pi^*|}\left(\Delta - \sqrt{\frac{2b^2}{t}\ln \frac{2}{\delta}}\right)} \right)^2$, with probability at least $1 - \delta$, it holds that $q_{t,i^*} \geq \tau$.*

*Proof of Lemma 8.* W.l.o.g, we assume $\mu_1 \leq \mu_2 \leq \ldots \mu_{n+k}$. Recall that we define $\Delta = \min_{i \neq i^*} \Delta_i = \mu_2 - \mu_1 = \frac{\mathbb{E}[\widetilde{L}_{t,2} - \widetilde{L}_{t,1}]}{t}$, and $\pi_1$ is the policy with the minimal expected loss.

Define

$$\delta_t \triangleq \widetilde{\ell}_{t-1,i'} - \widetilde{\ell}_{t-1,1}. \tag{7}$$

where $i' \triangleq \arg\min_{i \neq 1} \widetilde{L}_{t-1,i}$ denotes the index of the best empirical policy up to $t-1$ other than $\pi_1$. Therefore for $i \geq 2$, it holds that

$$\widetilde{L}_{t-1,i'} - \widetilde{L}_{t-1,i} = \sum_{s=1}^{t-1} \delta_s \leq 0.$$

We have $q_{t,i^*} = q_{t,1} = \frac{\exp(-\eta_t \widetilde{L}_{t-1,1})}{\sum_{i=1}^{|\Pi^*|} \exp(-\eta_t \widetilde{L}_{t-1,i})}$ as the weight of optimal expert at round $t$. Therefore

$$
\begin{aligned}
q_{t,i^*} = q_{t,1} &= \frac{\exp\left(-\eta_t \widetilde{L}_{t-1,1}\right)}{\sum_{i=1}^{|\Pi^*|} \exp\left(-\eta_t \widetilde{L}_{t-1,i}\right)} \\
&\overset{(a)}{=} \frac{\exp\left(-\eta_t \widetilde{L}_{t-1,1} + \eta_t \widetilde{L}_{t-1,i'}\right)}{\sum_{i=1}^{|\Pi^*|} \exp\left(-\eta_t \widetilde{L}_{t-1,i} + \eta_t \widetilde{L}_{t-1,i'}\right)} \\
&\overset{(b)}{=} \frac{\exp\left(\eta_t \sum_{s=1}^{t} \delta_s\right)}{\exp\left(\eta_t \sum_{s=1}^{t} \delta_s\right) + \sum_{i=2}^{|\Pi^*|} \exp\left(-\eta_t \widetilde{L}_{t-1,i} + \eta_t \widetilde{L}_{t-1,i'}\right)} \\
&\geq \frac{\exp\left(\eta_t \sum_{s=1}^{t} \delta_s\right)}{\exp\left(\eta_t \sum_{s=1}^{t} \delta_s\right) + |\Pi^*|-1}
\end{aligned}
\tag{8}
$$

where step $(a)$ is by dividing the cumulative loss of sub-optimal policy $\pi_{i'}$ and step (b) is by the definition of $\delta_t$ in Equation (7).

---

[7]We assume $p_{\min} > 0$ per the policy regularization criterion in Appendix C.3. (cf. Algorithm 1 on "Regularized policy $\bar{\pi}(\mathbf{x}_t)$)".

Let $\tau \in (0,1)$, such that $q_{t,i^*} \geq \frac{\exp\left(\eta_t \sum_{s=1}^t \delta_s\right)}{\exp\left(\eta_t \sum_{s=1}^t \delta_s\right) + |\Pi^*| - 1} \geq \tau$. Plugging in $\eta_t = \sqrt{\frac{\ln |\Pi^*|}{t}}$ and define $\overline{\delta}_t = \frac{1}{t} \sum_{s=1}^t \delta_s$, we get

$$\frac{\exp\left(\sqrt{\ln |\Pi^*|} \sqrt{t} \cdot \overline{\delta}_t\right)}{\exp\left(\sqrt{\ln |\Pi^*|} \sqrt{t} \cdot \overline{\delta}_t\right) + |\Pi^*| - 1} \geq \tau$$

Therefore, we obtain $\exp\left(\sqrt{\ln |\Pi^*|} \sqrt{t} \cdot \overline{\delta}_t\right) \geq \frac{(|\Pi^*| - 1)\tau}{1 - \tau}$. Rearranging the terms, we get

$$t \geq \left(\frac{\ln \frac{(|\Pi^*| - 1)\tau}{1 - \tau}}{\sqrt{\ln |\Pi^*|} \cdot \overline{\delta}_t}\right)^2$$

Next, we seek a high probability upper bound on $\overline{\delta}_t$. Denote $\Delta_i \triangleq \mu_i - \mu_1$ for $i \in 1, \ldots, |\Pi^*|$. We know

$$P(\overline{\delta}_t \leq \Delta_2 - \epsilon) \overset{(a)}{\leq} P(\overline{\delta}_t \leq \Delta_{i'} - \epsilon) = P(\frac{1}{t} \sum_{s=1}^t \delta_s - \Delta_{i'} \leq -\epsilon) \overset{(b)}{\leq} e^{-\frac{t\epsilon^2}{2b^2}} \tag{9}$$

Here, step (9a) is by the fact that $\Delta_2 = \min_{i \neq 1} \Delta_i \leq \Delta_{i'}$, and step (9b) is by Hoeffding's inequality where $b$ denotes the upper bound on $|\delta_s|$. Further note that

$$\delta_{s+1} = \tilde{\ell}_{s,i'} - \tilde{\ell}_{s,1} = \frac{U_s}{z_s} \langle \pi_{i'}(\mathbf{x}_s) - \pi_1(\mathbf{x}_s), \mathbb{I}\{\hat{\mathbf{y}}_s \neq y_s\}\rangle$$

$$\leq \frac{\langle \pi_{i'}(\mathbf{x}_s), \mathbb{I}\{\hat{\mathbf{y}}_s \neq y_s\}\rangle}{z_s}$$

$$\overset{\text{Eq. (4)}}{\leq} U_s \frac{\langle \pi_{i'}(\mathbf{x}_s), \mathbb{I}\{\hat{\mathbf{y}}_s \neq y_s\}\rangle}{\frac{1}{c} \sum_{y \in \mathcal{Y}} \langle \boldsymbol{w}_s, \mathbb{I}\{\hat{\mathbf{y}}_s \neq y\}\rangle \log_c \frac{1}{\langle \boldsymbol{w}_s, \mathbb{I}\{\hat{\mathbf{y}}_s \neq y\}\rangle}}$$

Given $p_{\min} = \min_{s,i} \pi(\mathbf{x}_s)$, we obtain $\delta_{s+1} \leq \frac{1}{p_{\min} \log_c(1/p_{\min})}$ and similarly, $\delta_{s+1} \geq -\frac{\langle \pi_1(\mathbf{x}_s), \mathbb{I}\{\hat{\mathbf{y}}_s \neq y_s\}\rangle}{z_s} \geq -\frac{1}{p_{\min} \log_c(1/p_{\min})}$. We hence conclude that $|\delta_{s+1}| \leq b$.

Let $2e^{-\frac{t\epsilon^2}{2b^2}} = \delta$. Therefore, when $t \geq \left(\frac{\ln \frac{(|\Pi^*| - 1)\tau}{1 - \tau}}{\sqrt{\ln |\Pi^*|}(\Delta - \epsilon)}\right)^2 = \left(\frac{\ln \frac{(|\Pi^*| - 1)\tau}{1 - \tau}}{\sqrt{\ln |\Pi^*|}\left(\Delta - \sqrt{\frac{2b^2}{t} \ln \frac{2}{\delta}}\right)}\right)^2$, it holds that $q_{t,i^*} \geq \tau$ with probability at least $1 - \delta$.

$\square$

**Lemma 9.** *At round t, when $t \geq \left(\frac{\ln \frac{|\Pi^*| - 1}{\gamma} + \sqrt{\ln |\Pi^*| \cdot 2b^2 \ln \frac{2}{\delta}}}{\sqrt{\ln |\Pi^*|}\Delta}\right)^2$, it holds that the arm chosen by the best policy $i^*$ will be the arm chosen by Algorithm 1 with probability at least $1 - \delta$. That is, $\arg\max\left\{\sum_{i \in [|\Pi^*|]} q_{t,i} \pi_i(\mathbf{x}_t)\right\} = \arg\max\{\pi_{i^*}(\mathbf{x}_t)\}$.*

*Proof of Lemma 9.* At round t, for Algorithm 1, we have loss $\sum_{j=1}^k \mathbb{I}\left\{j = \arg\max\left\{\sum_{i \in [|\Pi^*|]} q_{t,i} \pi_i(\boldsymbol{x}_t)\right\}\right\} \widehat{\ell}_{t,j}$. Let $q_{t,i^*} \geq \tau$. At round t, the best policy $i^*$'s top weight arm $j_{t,i^*}$'s probability $\max\{\pi_{i^*}(\boldsymbol{x}_t)\}$ is at least $\frac{1}{k}$. The second rank probability of $\pi_{i^*}(\boldsymbol{x}_t)$ is $\max_j [\pi_{i^*}(\boldsymbol{x}_t)]_{j \neq \text{maxind}(\pi_{i^*}(\boldsymbol{x}_t))}$. Let us define

$$\gamma := \min_{\boldsymbol{x}_t} \left\{\max_{w_j \in \boldsymbol{w}_{i^*}^t} w_j - \max_{w_j \in \boldsymbol{w}_{i^*}^t, j \neq \text{maxind}(\boldsymbol{w}_{i^*}^t)} w_j\right\} \tag{10}$$

$$= \max\{\pi_{i^*}(\boldsymbol{x}_t)\} - \max_j \left\{[\pi_{i^*}(\boldsymbol{x}_t)]_{j \neq \text{maxind}(\pi_{i^*}(\boldsymbol{x}_t))}\right\},$$

as the minimal gap in model distribution space of best policy. The arm recommended by the best policy $i^*$ of CAMS will dominate CAMS's selection, when we have

$$q_{t,i^*} \cdot \max\{\pi_{i^*}(\boldsymbol{x}_t)\} \geq (1 - q_{t,i^*}) + q_{t,i^*}\left(\max_j [\pi_{i^*}(\boldsymbol{x}_t)]_{j \neq \text{maxind}(\pi_{i^*}(\boldsymbol{x}_t))}\right) \tag{11}$$

Rearranging the terms, and by

$$q_{t,i^*} \cdot \gamma \overset{\text{Eq. (10)}}{=} q_{t,i^*}\left(\max\{\pi_{i^*}(\boldsymbol{x}_t)\} - \max_j [\pi_{i^*}(\boldsymbol{x}_t)]_{j \neq \text{maxind}(\pi_{i^*}(\boldsymbol{x}_t))}\right) \geq (1 - q_{t,i^*})$$

Therefore, we get $\tau \cdot (\gamma) \geq (1 - \tau)$, and thus $\tau \geq \frac{1}{\gamma+1}$.

Set $\tau \geq \frac{1}{\gamma+1}$. By Lemma 8, we get

$$t \geq \left(\frac{\ln \frac{|\Pi^*-1|\tau}{1-\tau}}{\sqrt{\ln |\Pi^*|}(\Delta - \epsilon)}\right)^2$$

$$\geq \left(\frac{\ln\left(\frac{|\Pi^*|-1}{\gamma}\right)}{\sqrt{\ln |\Pi^*|}(\Delta - \epsilon)}\right)^2$$

$$\overset{(c)}{\geq} \left(\frac{\ln \frac{|\Pi^*|-1}{\gamma}}{\sqrt{\ln |\Pi^*|}\Delta - \sqrt{\ln |\Pi^*| \cdot \frac{2b^2}{t} \ln \frac{2}{\delta}}}\right)^2$$

where the last step is by applying $2e^{-\frac{t\epsilon^2}{2b^2}} = \delta$, thus, $\epsilon = \sqrt{\frac{2b^2}{t} \ln \frac{2}{\delta}}$. Dividing both sides by $t$

$$1 \overset{(d)}{\geq} \left(\frac{\ln \frac{|\Pi^*|-1}{\gamma}}{\sqrt{\ln |\Pi^*| \cdot t}\Delta - \sqrt{\ln |\Pi^*| \cdot 2b^2 \ln \frac{2}{\delta}}}\right)^2$$

$$\ln \frac{|\Pi^*|-1}{\gamma} \leq \sqrt{t}\sqrt{\ln(|\Pi^*|)}\Delta - \sqrt{\ln(|\Pi^*|) \cdot 2b^2 \ln \frac{2}{\delta}}$$

$$t \geq \left(\frac{\ln \frac{|\Pi^*|-1}{\gamma} + \sqrt{\ln |\Pi^*| \cdot 2b^2 \ln \frac{2}{\delta}}}{\sqrt{\ln |\Pi^*|}\Delta}\right)^2.$$

So, when $t \geq \left(\frac{\ln \frac{|\Pi^*|-1}{\gamma} + \sqrt{\ln |\Pi^*| \cdot 2b^2 \ln \frac{2}{\delta}}}{\sqrt{\ln |\Pi^*|}\Delta}\right)^2$, it holds that $\arg\max\left\{\sum_{i \in [|\Pi^*|]} q_{t,i}\pi_i(\boldsymbol{x}_t)\right\} = \arg\max\{\pi_{i^*}(\boldsymbol{x}_t)\}$. $\square$

*Proof of Theorem 1.* Therefore, with probability at least $1 - \delta$, we get constant regret $\left(\frac{\ln \frac{|\Pi^*|-1}{\gamma} + \sqrt{\ln |\Pi^*| \cdot 2b^2 \ln \frac{2}{\delta}}}{\sqrt{\ln |\Pi^*|}\Delta}\right)^2$.

Furthermore, with probability at most $\delta$, the regret is upper bounded by $T$. Thus, we have

$$\overline{\mathcal{R}}(T) \leq (1 - \delta)\left(\frac{\ln \frac{|\Pi^*|-1}{\gamma} + \sqrt{\ln |\Pi^*| \cdot 2b^2 \ln \frac{2}{\delta}}}{\sqrt{\ln |\Pi^*|}\Delta}\right)^2 + \delta T$$

$$\overset{(a)}{\leq} \left(1 - \frac{1}{T}\right)\left(\frac{\ln \frac{|\Pi^*|-1}{\gamma} + b\sqrt{\ln |\Pi^*| \cdot (2\ln T + 2\ln 2)}}{\sqrt{\ln |\Pi^*|}\Delta}\right)^2 + 1$$

$$= O\left(\frac{b \ln T}{\Delta^2} + \left(\frac{\ln \frac{|\Pi^*|-1}{\gamma}}{\sqrt{\ln |\Pi^*|}\Delta}\right)^2\right),$$

where step (a) by setting $\delta = \frac{1}{T}$, and where $\gamma$ in Eq. (10) is the min gap. $\qquad\square$

### E.2 Proof of Theorem 2

In this section, we analyze the query complexity of CAMS in the stochastic setting, where we take a similar approach as proposed by Karimi et al. [39] for the context-free model selection problem. Our main idea is to derive from query indicator $U_t$ and query probability $z_t$. We first used Lemma 10 to bound the expected number of queries $\sum_{t=1}^{T} U_t$ by the sum of query probability as $\sum_{t=1}^{T} \delta_0^t + \sum_{t=1}^{T} \mathfrak{E}(\hat{\mathbf{y}}_t, \mathbf{w}_t)$. Then we used Lemma 11 to bound the first item (which corresponds to the lower bound of query probability over $T$ rounds) and applied Lemma 12 to bound the second term (which characterizes the model disagreement). Finally, we combined the upper bounds on the two parts to reach the desired result.

**Lemma 10.** *The query complexity of Algorithm 1 is upper bounded by*

$$\mathbb{E}\left[\sum_{t=1}^{T}\left(\frac{1}{\sqrt{t}} + \frac{\sum_{y\in\mathcal{Y}}\langle\mathbf{w}_t, \boldsymbol{\ell}_t^y\rangle \log_{|\mathcal{Y}|}\frac{1}{\langle\mathbf{w}_t, \boldsymbol{\ell}_t^y\rangle}}{|\mathcal{Y}|}\right)\right]. \tag{12}$$

*Proof.* Now we have model disagreement defined in Eq. (3), the query probability defined in Eq. (4), and the query indicator $U$. Let us assume, at each round, we have query probability $z_t > 0$, which indicates we will not process the instance that all the models' prediction are the same.

At round $t$, from query probability Eq. (4), we have

$$z_t = \max\left\{\delta_0^t, \mathfrak{E}(\hat{\mathbf{y}}_t, \mathbf{w}_t)\right\}$$
$$\leq \delta_0^t + \mathfrak{E}(\hat{\mathbf{y}}_t, \mathbf{w}_t),$$

where the inequality is by applying that $\forall A, B \geq 0, \max\{A, B\} \leq A + B$.

Thus, in total round $T$, we could get the following equation as the cumulative query cost,

$$\mathbb{E}\left[\sum_{t=1}^{T} U_t\right] \leq \mathbb{E}\left[\sum_{t=1}^{T}\left(\frac{1}{\sqrt{t}} + \frac{\sum_{y\in\mathcal{Y}}\langle\mathbf{w}_t, \boldsymbol{\ell}_t^y\rangle \log_{|\mathcal{Y}|}\frac{1}{\langle\mathbf{w}_t, \boldsymbol{\ell}_t^y\rangle}}{|\mathcal{Y}|}\right)\right], \tag{13}$$

where the inequality is by inputting $\delta_0^t = \frac{1}{\sqrt{t}}$ and Eq. (3). $\qquad\square$

**Lemma 11.** $\sum_{t=1}^{T} \frac{1}{\sqrt{t}} \leq 2\sqrt{T}.$

*Proof.* We can bound the LHS as follows:

$$\sum_{t=1}^{T} \frac{1}{\sqrt{t}} = \sum_{t=1}^{\lfloor\sqrt{T}\rfloor} \frac{1}{\sqrt{t}} + \sum_{t=\lfloor\sqrt{T}\rfloor+1}^{T} \frac{1}{\sqrt{t}}$$
$$\leq \sqrt{T} + \sum_{t=\lfloor\sqrt{T}\rfloor+1}^{T} \frac{1}{\sqrt{T}}$$
$$= \sqrt{T} + \left(T - \sqrt{T}\right)\frac{1}{\sqrt{T}}$$
$$\leq 2\sqrt{T}.$$

$\qquad\square$

**Lemma 12.** *Denote the true label at round $t$ by $y_t$, and define $p_{t,y} := \sum_{j\in[k]} \mathbb{I}\{\hat{y}_{t,j} = y\} w_j$. Further define $R_t := \sum_t 1 - p_{t,y_t}$ as the expected cumulative loss of Algorithm 1 at $t$. Then*

$$\sum_{t=1}^{T} \frac{\sum_{y\in\mathcal{Y}}\langle\mathbf{w}_t, \boldsymbol{\ell}_t^y\rangle \log_{|\mathcal{Y}|}\frac{1}{\langle\mathbf{w}_t, \boldsymbol{\ell}_t^y\rangle}}{|\mathcal{Y}|} \leq \frac{R_T \cdot \left(\log_{|\mathcal{Y}|}\frac{T^2(|\mathcal{Y}|-1)}{R_T^2}\right)}{|\mathcal{Y}|}.$$

*Proof of Lemma 12.* Suppose at round $t$, the true label is $y_t$. $\sum_{y \neq y_t} p_{t,y} = 1 - p_{t,y_t} = 1 - \left\langle \sum_{i \in |\Pi^*|} q_{t,i} \pi_i(\boldsymbol{x}_t), \boldsymbol{\ell}_t \right\rangle = r_t,$

$$
\begin{aligned}
\frac{\sum_{y \in \mathcal{Y}} \langle \boldsymbol{w}_t, \boldsymbol{\ell}_t^y \rangle \log_{|\mathcal{Y}|} \frac{1}{\langle \boldsymbol{w}_t, \boldsymbol{\ell}_t^y \rangle}}{|\mathcal{Y}|} &= \frac{(1 - p_{t,y_t}) \log_{|\mathcal{Y}|} \frac{1}{1 - p_{t,y_t}}}{|\mathcal{Y}|} + \frac{\sum_{y \neq y_t} (1 - p_{t,y}) \log_{|\mathcal{Y}|} \frac{1}{1 - p_{t,y}}}{|\mathcal{Y}|} \\
&\overset{(a)}{\leq} \frac{(1 - p_{t,y_t}) \log_{|\mathcal{Y}|} \frac{1}{1 - p_{t,y_t}}}{|\mathcal{Y}|} + (|\mathcal{Y}| - 1) \frac{\frac{(1 - p_{t,y_t})}{|\mathcal{Y}| - 1} \log_{|\mathcal{Y}|} \frac{|\mathcal{Y}| - 1}{1 - p_{t,y_t}}}{|\mathcal{Y}|} \\
&\leq \frac{(1 - p_{t,y_t}) \log_{|\mathcal{Y}|} \frac{1}{1 - p_{t,y_t}}}{|\mathcal{Y}|} + \frac{(1 - p_{t,y_t}) \log_{|\mathcal{Y}|} \frac{|\mathcal{Y}| - 1}{1 - p_{t,y_t}}}{|\mathcal{Y}|} \\
&= \frac{(1 - p_{t,y_t}) \log_{|\mathcal{Y}|} \frac{|\mathcal{Y}| - 1}{(1 - p_{t,y_t})^2}}{|\mathcal{Y}|} \\
&\overset{(b)}{\leq} \frac{r_t \log_{|\mathcal{Y}|} \frac{|\mathcal{Y}| - 1}{r_t^2}}{|\mathcal{Y}|},
\end{aligned}
$$

where step $(a)$ is by applying Jensen's inequality and using $1 - p_{t,y} = \frac{1 - p_{t,y_t}}{|\mathcal{Y}| - 1}$, and step $(b)$ is by replacing the expected loss $1 - p_{t,y_t}$ by its short-hand notation $r_t$.

Recall that we define the expected cumulative loss as $R_T = \sum_{t=1}^T r_t$. Since when $r_t \in [0, 1]$, $\frac{r_t \log_{|\mathcal{Y}|} \frac{|\mathcal{Y}| - 1}{r_t^2}}{|\mathcal{Y}|}$ is concave, we get

$$
\sum_{t=1}^T \frac{\sum_{y \in \mathcal{Y}} \langle \boldsymbol{w}_t, \boldsymbol{\ell}_t^y \rangle \log_{|\mathcal{Y}|} \frac{1}{\langle \boldsymbol{w}_t, \boldsymbol{\ell}_t^y \rangle}}{|\mathcal{Y}|} \leq \frac{T \left( \frac{\sum r_t}{T} \right) \left( \log_{|\mathcal{Y}|} \frac{|\mathcal{Y}| - 1}{\frac{\sum r_t}{T} \frac{\sum r_t}{T}} \right)}{|\mathcal{Y}|} = \frac{R_T \left( \log_{|\mathcal{Y}|} \frac{T^2 (|\mathcal{Y}| - 1)}{R_T^2} \right)}{|\mathcal{Y}|}.
\tag{14}
$$

Since $R_T$ is the cumulative loss up to round $T$, $T$'s incremental rate is no less than $R_T$'s incremental rate. Thus, $R_T \leq T$ and $\frac{T_t}{R_t} \leq \frac{T_{t+1}}{R_{t+1}}$. So we get Eq. (14). $\qquad \square$

Now we are ready to prove Theorem 2.

*Proof of Theorem 2.* From Lemma 10, we get the following equation as the cumulative query cost

$$
\mathbb{E} \left[ \sum_{t=1}^T U_t \right] \leq \mathbb{E} \left[ \sum_{t=1}^T \left( \frac{1}{\sqrt{t}} + \frac{\sum_{y \in \mathcal{Y}} \langle \boldsymbol{w}_t, \boldsymbol{\ell}_t^y \rangle \log_{|\mathcal{Y}|} \frac{1}{\langle \boldsymbol{w}_t, \boldsymbol{\ell}_t^y \rangle}}{|\mathcal{Y}|} \right) \right].
$$

Let us assume the expected total loss of best policy is $T \mu_{i^*}$. From Theorem 1, we get

$$
\mathbb{E} [R_T] = \mathbb{E} \left[ \sum_{t=1}^T r_t \right] \leq \left( \frac{\ln \frac{|\Pi^*| - 1}{\gamma} + \sqrt{\ln |\Pi^*| \cdot 2b^2 \ln \frac{2}{\delta}}}{\sqrt{\ln |\Pi^*|} \Delta} \right)^2 + T \mu_{i^*}.
$$

Plugging this result into the query complexity bound given by Lemma 11 and Lemma 12, we have

$$\mathbb{E}\left[\sum_{t=1}^{T} U_t\right] \leq 2\sqrt{T} + \frac{\left(\left(\frac{\ln\frac{|\Pi^*|-1}{\gamma}+\sqrt{\ln|\Pi^*|\cdot 2b^2\ln\frac{2}{\delta}}}{\sqrt{\ln|\Pi^*|\Delta}}\right)^2 + T\mu_{i^*}\right)}{|\mathcal{Y}|}\log_{|\mathcal{Y}|}\frac{T^2(|\mathcal{Y}|-1)}{\left(\left(\frac{\ln\frac{|\Pi^*|-1}{\gamma}+\sqrt{\ln|\Pi^*|\cdot 2b^2\ln\frac{2}{\delta}}}{\sqrt{\ln|\Pi^*|\Delta}}\right)^2 + T\mu_{i^*}\right)^2}$$

$$\leq \frac{\left(\left(\frac{\ln\frac{|\Pi^*|-1}{\gamma}+\sqrt{\ln|\Pi^*|\cdot 2b^2\ln\frac{2}{\delta}}}{\sqrt{\ln|\Pi^*|\Delta}}\right)^2 + T\mu_{i^*}\right)\left(\log_{|\mathcal{Y}|}(T|\mathcal{Y}|)\right)}{|\mathcal{Y}|}$$

$$= \frac{\left(\left(\frac{\ln\frac{|\Pi^*|-1}{\gamma}+\sqrt{\ln|\Pi^*|\cdot 2b^2\ln\frac{2}{\delta}}}{\sqrt{\ln|\Pi^*|\Delta}}\right)^2 + T\mu_{i^*}\right)\ln(T)}{|\mathcal{Y}|\ln|\mathcal{Y}|}$$

$$\stackrel{(a)}{=} \frac{\left(\left(\frac{\ln\frac{|\Pi^*|-1}{\gamma}+\sqrt{\ln|\Pi^*|\cdot 2b^2\ln\frac{2}{\delta}}}{\sqrt{\ln|\Pi^*|\Delta}}\right)^2 + T\mu_{i^*}\right)\ln(T)}{c\ln c},$$

where $\gamma$ is defined as Eq. (10) and step (a) by applying $c = |\mathcal{Y}|$. $\qquad\square$

## F  Proofs for the Adversarial Setting

In this section, we first prove the regret bound presented in Theorem 3 and then prove the query complexity bound presented in Theorem 4 for Algorithm 1 in the adversarial setting. Lemma 13 builds upon the proof of the hedge algorithm [27], but with an *adaptive* learning rate.

### F.1  Proof of Theorem 3

**Lemma 13.** *Consider the setting of Algorithm 1, Let us define* $h_{t,i} = \exp\left(-\eta_t \tilde{L}_{t-1,i}\right) \forall i \in |\Pi^*|$ *as exponential cumulative loss of policy* $i$, $\eta_t$ *is the adaptive learning rate and* $\mathbf{q}_t$ *is the probability distribution of policies, then*

$$\log\frac{\sum_{i\in[|\Pi^*|]}h_{T+1,i}}{\sum_{i\in[|\Pi^*|]}h_{1,i}} \leq -\sum_{t=1}^{T}\eta_t\sum_{i=1}^{|\Pi^*|}q_{t,i}\widetilde{\ell}_{t,i} + \sum_{t=1}^{T}\frac{\eta_t^2}{2}\sum_{i=1}^{|\Pi^*|}q_{t,i}\left(\widetilde{\ell}_{t,i}\right)^2.$$

*Proof.* We first bound the following term

$$\frac{\sum_{i\in[|\Pi^*|]}h_{t+1,i}}{\sum_{i\in[|\Pi^*|]}h_{t,i}} = \sum_{i=1}^{|\Pi^*|}\frac{h_{t+1,i}}{\sum_{i\in[|\Pi^*|]}h_{t,i}}$$

$$= \sum_{i=1}^{|\Pi^*|}q_{t,i}\exp\left(-\eta_t\widetilde{\ell}_{t,i}\right)$$

$$\leq \sum_{i=1}^{|\Pi^*|}q_{t,i}\left(1 - \eta_t\widetilde{\ell}_{t,i} + \frac{\eta_t^2\left(\widetilde{\ell}_{t,i}\right)^2}{2}\right)$$

$$= 1 - \eta_t\sum_{i=1}^{|\Pi^*|}q_{t,i}\widetilde{\ell}_{t,i} + \frac{\eta_t^2}{2}\sum_{i=1}^{|\Pi^*|}q_{t,i}\left(\widetilde{\ell}_{t,i}\right)^2,$$

where the inequality is by applying that for $x \leq 0$, we have $e^x \leq 1 + x + \frac{x^2}{2}$.

By taking $\log$ on both side, we get

$$\log \frac{\sum_{i \in [|\Pi^*|]} h_{t+1,i}}{\sum_{i \in [|\Pi^*|]} h_{t,i}} \leq \log \left(1 - \eta_t \sum_{i=1}^{|\Pi^*|} q_{t,i} \widetilde{\ell}_{t,i} + \frac{\eta_t^2}{2} \sum_{i=1}^{|\Pi^*|} q_{t,i} \left(\widetilde{\ell}_{t,i}\right)^2\right)$$

$$\overset{(a)}{\leq} -\eta_t \sum_{i=1}^{|\Pi^*|} q_{t,i} \widetilde{\ell}_{t,i} + \frac{\eta_t^2}{2} \sum_{i=1}^{|\Pi^*|} q_{t,i} \left(\widetilde{\ell}_{t,i}\right)^2,$$

where step $(a)$ is by applying that $\log (1 + x) \leq x$, when $x \geq -1$.

Now summing over $t = 1 : T$ yields:

$$\log \frac{\sum_{i \in [|\Pi^*|]} h_{T+1,i}}{\sum_{i \in [|\Pi^*|]} h_{1,i}} = \sum_{t=1}^{T} \log \frac{\sum_{i \in [|\Pi^*|]} h_{t+1,i}}{\sum_{i \in [|\Pi^*|]} h_{t,i}}$$

$$\leq -\sum_{t=1}^{T} \eta_t \sum_{i=1}^{|\Pi^*|} q_{t,i} \widetilde{\ell}_{t,i} + \sum_{t=1}^{T} \frac{\eta_t^2}{2} \sum_{i=1}^{|\Pi^*|} q_{t,i} \left(\widetilde{\ell}_{t,i}\right)^2.$$

$\square$

**Lemma 14.** *Consider the setting of Algorithm 1. Let $p_{t,y} = \sum_{j \in [k]} \mathbb{I}\{\hat{y}_{t,j} = y\} w_j$. The query probability $z_t$ satisfies*

$$z_t \geq \frac{1}{|\mathcal{Y}| \ln |\mathcal{Y}|} \left(p_{t,y_t} (1 - p_{t,y_t}) + p_{t,y} (1 - p_{t,y})\right), \forall y \neq y_t.$$

*Proof.* We first bound the query probability term

$$z_t = \max \{\delta_0^t, \mathfrak{E}(\hat{\mathbf{y}}_t, \mathbf{w}_t)\}$$

$$= \max\{\delta_0^t, \frac{1}{|\mathcal{Y}|} \sum_{y \in \mathcal{Y}} \langle \mathbf{w}_t, \boldsymbol{\ell}_t^y \rangle \log_{|\mathcal{Y}|} \frac{1}{\langle \mathbf{w}_t, \boldsymbol{\ell}_t^y \rangle}\}$$

$$= \max\{\delta_0^t, \frac{1}{|\mathcal{Y}|} \sum_{y \in \mathcal{Y}} (1 - p_{t,y}) \cdot \ln \frac{1}{1 - p_{t,y}} \frac{1}{\ln |\mathcal{Y}|}\}$$

$$\overset{(a)}{\geq} \max\{\delta_0^t, \frac{1}{|\mathcal{Y}|} \sum_{y \in \mathcal{Y}} (1 - p_{t,y}) \cdot p_{t,y} \cdot \frac{1}{\ln |\mathcal{Y}|}\}$$

$$= \max\{\delta_0^t, \frac{1}{|\mathcal{Y}| \ln |\mathcal{Y}|} \sum_{y \in \mathcal{Y}} (1 - p_{t,y}) \cdot p_{t,y}\}$$

$$\overset{(b)}{\geq} \frac{1}{|\mathcal{Y}| \ln |\mathcal{Y}|} \left(p_{t,y_t} (1 - p_{t,y_t}) + p_{t,y} (1 - p_{t,y})\right), \forall y \neq y_t,$$

where step $(a)$ is by applying $\ln (1 + x) \geq \frac{x}{1+x}$ for $x > -1$,

$$\ln \frac{1}{1 - p_{t,y}} = \ln \left(1 + \frac{p_{t,y}}{1 - p_{t,y}}\right) \geq \frac{\frac{p_{t,y}}{1 - p_{t,y}}}{\frac{1}{1 - p_{t,y}}} = p_{t,y},$$

and where step $(b)$ is by applying $\forall a, b \in \mathbb{R}, \max \{a, b\} \geq a$.

$\square$

*Proof of Theorem 3.* By applying Lemma 13, we got

$$\log \frac{\sum_{i \in [|\Pi^*|]} h_{T+1,i}}{\sum_{i \in [|\Pi^*|]} h_{1,i}} \leq -\sum_{t=1}^{T} \eta_t \sum_{i=1}^{|\Pi^*|} q_{t,i} \widetilde{\ell}_{t,i} + \sum_{t=1}^{T} \frac{\eta_t^2}{2} \sum_{i=1}^{|\Pi^*|} q_{t,i} \left(\widetilde{\ell}_{t,i}\right)^2.$$

For any policy $s$, we have a lower bound

$$\log \frac{\sum_{i \in [|\Pi^*|]} h_{T+1,i}}{\sum_{i \in [|\Pi^*|]} h_{1,i}} \geq \log \frac{h_{T+1,s}}{\sum_{i \in [|\Pi^*|]} h_{1,i}}$$

$$\overset{(a)}{=} \log \frac{h_{T+1,s}}{|\Pi^*|}$$

$$= -\log(n+k) - \eta_T \sum_{t=1}^{T} \widetilde{\ell}_{t,s}, \tag{15}$$

where step $(a)$ in Eq. (15) is by initializing $\widetilde{L}_0 = 0$, $e^0 = 1$, and $\sum_{i \in [|\Pi^*|]} \boldsymbol{h}_1 = e^{(-\eta_t \widetilde{L}_0)} = |\Pi^*|$.

Thus, we have

$$-\sum_{t=1}^{T} \eta_t \sum_{i=1}^{|\Pi^*|} q_{t,i} \widetilde{\ell}_{t,i} + \sum_{t=1}^{T} \frac{\eta_t^2}{2} \sum_{i=1}^{|\Pi^*|} q_{t,i} \left(\widetilde{\ell}_{t,i}\right)^2 \geq -\log(n+k) - \eta_T \sum_{t=1}^{T} \widetilde{\ell}_{t,s}$$

$$\sum_{t=1}^{T} \eta_t \sum_{i=1}^{|\Pi^*|} q_{t,i} \widetilde{\ell}_{t,i} - \eta_T \sum_{t=1}^{T} \widetilde{\ell}_{t,s} \leq \log(n+k) + \sum_{t=1}^{T} \frac{\eta_t^2}{2} \sum_{i=1}^{|\Pi^*|} q_{t,i} \left(\widetilde{\ell}_{t,i}\right)^2$$

$$\eta_T \sum_{t=1}^{T} \sum_{i=1}^{|\Pi^*|} q_{t,i} \widetilde{\ell}_{t,i} - \eta_T \sum_{t=1}^{T} \widetilde{\ell}_{t,s} \overset{(b)}{\leq} \log(n+k) + \sum_{t=1}^{T} \frac{\eta_t^2}{2} \sum_{i=1}^{|\Pi^*|} q_{t,i} \left(\widetilde{\ell}_{t,i}\right)^2$$

$$\sum_{t=1}^{T} \sum_{i=1}^{|\Pi^*|} q_{t,i} \widetilde{\ell}_{t,i} - \sum_{t=1}^{T} \widetilde{\ell}_{t,s} \overset{(c)}{\leq} \frac{\log|\Pi^*|}{\eta_T} + \frac{1}{\eta_T} \sum_{t=1}^{T} \frac{\eta_t^2}{2} \sum_{i=1}^{|\Pi^*|} q_{t,i} \left(\widetilde{\ell}_{t,i}\right)^2,$$

where step $(b)$ is by applying

$$\eta_T \sum_{t=1}^{T} \sum_{i=1}^{|\Pi^*|} q_{t,i} \widetilde{\ell}_{t,i} - \eta_T \sum_{t=1}^{T} \widetilde{\ell}_{t,s} \leq \sum_{t=1}^{T} \eta_t \sum_{i=1}^{|\Pi^*|} q_{t,i} \widetilde{\ell}_{t,i} - \eta_T \sum_{t=1}^{T} \widetilde{\ell}_{t,s},$$

and step $(c)$ is by dividing $\eta_T$ on both side.

Because we have

$$\mathbb{E}_T \left[ q_{t,i} \left(\widetilde{\ell}_{t,i}\right)^2 \right] = q_{t,i} \mathbb{E}_T \left[ \left(\pi_i(\boldsymbol{x}_t) \cdot \widehat{\boldsymbol{\ell}}_t\right)^2 \right]$$

$$= q_{t,i} \left( P(U_t = 1) \left(\pi_i(\boldsymbol{x}_t) \cdot \frac{\boldsymbol{\ell}_t}{z_t}\right)^2 + P(U_t = 0) \cdot 0 \right)$$

$$= q_{t,i} \left( z_t \left(\pi_i(\boldsymbol{x}_t) \cdot \frac{\boldsymbol{\ell}_t}{z_t}\right)^2 \right)$$

$$= \frac{q_{t,i}}{z_t} (\pi_i(\boldsymbol{x}_t) \cdot \boldsymbol{\ell}_t)^2$$

$$\leq \frac{q_{t,i}}{z_t} \pi_i(\boldsymbol{x}_t) \cdot \boldsymbol{\ell}_t$$

$$= \frac{q_{t,i}}{z_t} \langle \pi_i(\boldsymbol{x}_t), \boldsymbol{\ell}_t \rangle,$$

it leads to

$$\sum_{t=1}^{T}\sum_{i=1}^{|\Pi^*|} q_{t,i}\widetilde{\ell}_{t,i} - \sum_{t=1}^{T}\widetilde{\ell}_{t,s} \le \frac{\log|\Pi^*|}{\eta_T} + \frac{1}{\eta_T}\sum_{t=1}^{T}\frac{\eta_t^2}{2}\sum_{i=1}^{|\Pi^*|}\frac{q_{t,i}}{z_t}\langle\pi_i(\boldsymbol{x}_t),\boldsymbol{\ell}_t\rangle$$

$$\overset{(d)}{\le} \frac{\log|\Pi^*|}{\eta_T} + \frac{1}{\eta_T}\sum_{t=1}^{T}\frac{\eta_t^2}{2}\frac{\langle\boldsymbol{w}_t,\boldsymbol{\ell}_t\rangle}{z_t},$$

where step $(d)$ is by applying $\sum_{i=1}^{|\Pi^*|} q_{t,i}\langle\pi_i(\boldsymbol{x}_t),\boldsymbol{\ell}_t\rangle = \langle\boldsymbol{w}_t,\boldsymbol{\ell}_t\rangle$.

So we have,

$$\sum_{t=1}^{T}\sum_{i=1}^{|\Pi^*|} q_{t,i}\widetilde{\ell}_{t,i} - \sum_{t=1}^{T}\widetilde{\ell}_{t,s} \le \frac{\log|\Pi^*|}{\eta_T} + \frac{1}{\eta_T}\sum_{t=1}^{T}\frac{\eta_t^2}{2}\frac{\langle\boldsymbol{w}_t,\boldsymbol{\ell}_t\rangle}{z_t}$$

$$\overset{(e)}{\le} \frac{\log|\Pi^*|}{\eta_T} + \frac{1}{\eta_T}\sum_{t=1}^{T}\frac{\eta_t^2}{2}\frac{1-p_{t,y_t}}{z_t}$$

$$\overset{(f)}{\le} \frac{\log|\Pi^*|}{\eta_T} + \frac{1}{\eta_T}\sum_{t=1}^{T}\frac{\eta_t^2}{2}\frac{1-p_{t,y_t}}{\mathcal{Y}_0\left((1-p_{t,y_t})p_{t,y_t}+(1-p_{t,y})p_{t,y}\right)}$$

$$\le \frac{\log|\Pi^*|}{\eta_T} + \frac{1}{\eta_T}\sum_{t=1}^{T}\frac{\eta_t^2}{2}\frac{1}{\mathcal{Y}_0\left(p_{t,y_t}+\frac{1-p_{t,y}}{1-p_{t,y_t}}p_{t,y}\right)},$$

where step $(e)$ is by using $\langle\boldsymbol{w}_t,\boldsymbol{\ell}_t\rangle = 1 - p_{t,y_t}$ and step $(f)$ by using Lemma [14] and get lower bound of $z_t$ as $\frac{1}{|\mathcal{Y}|\ln|\mathcal{Y}|}\left(p_{t,y_t}(1-p_{t,y_t})+p_{t,y}(1-p_{t,y})\right)$ and applying $\frac{1}{|\mathcal{Y}|\ln|\mathcal{Y}|} = \mathcal{Y}_0$.

If $p_{t,y_t} \ge \frac{1}{|\mathcal{Y}|}$,

$$p_{t,y_t} + \frac{1-p_{t,y}}{1-p_{t,y_t}}p_{t,y} \ge \frac{1}{|\mathcal{Y}|}.$$

If $p_{t,y_t} < \frac{1}{|\mathcal{Y}|}$, $\exists y, p_{t,y} \to 1$, $\delta_1^t = 1 - \max_{y,\tau\in[t]}p_{\tau,y}$. Let $p_{t,\hat{y}} = \max_y p_{t,y}$. Thus, we have $w_{\hat{y}} > \frac{1}{|\mathcal{Y}|}$ and

$$p_{t,y_t} + \frac{1-p_{t,y}}{1-p_{t,y_t}}p_{t,y} \ge p_{t,y_t} + w_{\hat{y}}\frac{\delta_1^t}{1-p_{t,y_t}} \ge 0 + \frac{1}{|\mathcal{Y}|}\frac{\delta_1^t}{1} = \frac{\delta_1^t}{|\mathcal{Y}|}.$$

Therefore

$$\max\{p_{t,y_t} + p_{t,y}\frac{1-p_{t,y}}{1-p_{t,y_t}}\} = \begin{cases} \frac{1}{|\mathcal{Y}|} & \text{if } p_{t,y_t} \ge \frac{1}{|\mathcal{Y}|}, \\ \frac{\delta_1^t}{|\mathcal{Y}|} & \text{if } p_{t,y_t} < \frac{1}{|\mathcal{Y}|}. \end{cases}$$

So we have

$$\sum_{t=1}^{T}\sum_{i=1}^{|\Pi^*|}q_{t,i}\widetilde{\ell}_{t,i}-\sum_{t=1}^{T}\widetilde{\ell}_{t,s} \le \frac{\log|\Pi^*|}{\eta_T}+\frac{1}{\eta_T}\sum_{t=1}^{T}\frac{\eta_t^2}{2}\frac{1}{\mathcal{Y}_0\left(p_{t,y_t}+\frac{1-\boldsymbol{w}_y}{1-p_{t,y_t}}p_{t,y}\right)}$$

$$\overset{(g)}{\le}\frac{\log|\Pi^*|}{\eta_T}+\frac{1}{\eta_T}\sum_{t=1}^{T}\frac{\eta_t^2}{2}\frac{1}{\max\{\mathcal{Y}_0\frac{\delta_1^t}{|\mathcal{Y}|},\delta_0^t\}}$$

$$=\frac{\log|\Pi^*|}{\eta_T}+\frac{1}{\eta_T}\sum_{t=1}^{T}\frac{\eta_t^2}{2}\frac{|\mathcal{Y}|^2\ln|\mathcal{Y}|}{\max\{\delta_1^t,\delta_0^t|\mathcal{Y}|^2\ln|\mathcal{Y}|\}}$$

$$\overset{(h)}{\le}\frac{\log|\Pi^*|}{\eta_T}+\frac{1}{\eta_T}\sum_{t=1}^{T}\frac{\eta_t^2}{2}\cdot\frac{|\mathcal{Y}|^2\ln|\mathcal{Y}|}{\frac{\delta_1^t+\delta_0^t|\mathcal{Y}|^2\ln|\mathcal{Y}|}{2}}$$

$$=\frac{\log|\Pi^*|}{\eta_T}+\frac{1}{\eta_T}\sum_{t=1}^{T}\eta_t^2\frac{1}{\delta_1^t+\delta_0^t|\mathcal{Y}|^2\ln|\mathcal{Y}|}\cdot|\mathcal{Y}|^2\ln|\mathcal{Y}|,$$

where step $(g)$ is by getting the lower bound of $z_t$ as $\frac{\delta_1^t}{|\mathcal{Y}|}\le\frac{1}{|\mathcal{Y}|}$, $\delta_0^t\le\frac{\delta_0^t}{1-p_{t,y_t}}$ and step $(h)$ is by applying $\max\{A,B\}\ge\frac{A+B}{2}$.

Let us define $\rho_t\triangleq\min_{\tau\in[t]}\delta_1^\tau=1-\max_{c,\tau\in[t]}p_{t,y}^\tau$. We get

$$\mathbb{E}_T[\mathcal{R}_T]\le\frac{\log|\Pi^*|}{\eta_T}+\frac{1}{\eta_T}\sum_{t=1}^{T}\log|\Pi^*|\cdot\frac{1}{T}\le\frac{2\log|\Pi^*|}{\eta_T}$$

Let $\eta_t=\sqrt{\frac{\rho_t+\delta_0^t|\mathcal{Y}|^2\ln|\mathcal{Y}|}{|\mathcal{Y}|^2\ln|\mathcal{Y}|}}\cdot\sqrt{\frac{\log|\Pi^*|}{T}}$, we obtain

$$\mathbb{E}_T[\mathcal{R}_T]\le\frac{2\sqrt{\log|\Pi^*|}\cdot\sqrt{T}\cdot\sqrt{|\mathcal{Y}|^2\ln|\mathcal{Y}|}}{\sqrt{\rho_T+\delta_0^T|\mathcal{Y}|^2\ln|\mathcal{Y}|}}$$

$$\le2|\mathcal{Y}|\sqrt{\frac{T\ln|\mathcal{Y}|\log|\Pi^*|}{\max\{\rho_T,\sqrt{1/T}\}}}$$

where the last inequality is due to the fact that

$$\rho_T+\delta_0^T|\mathcal{Y}|^2\ln|\mathcal{Y}|>\max\{\rho_T,\delta_0^T\}=\max\{\rho_T,\sqrt{1/T}\}$$

which completes the proof. □

### F.2 Proof of Theorem 4

*Proof of Theorem 4.* From Lemma 10, we get the following equation as the cumulative query cost

$$\mathbb{E}\left[\sum_{t=1}^{T}U_t\right]\le\mathbb{E}\left[\sum_{t=1}^{T}\left(\frac{1}{\sqrt{t}}+\frac{\sum_{y\in\mathcal{Y}}\langle\boldsymbol{w}_t,\boldsymbol{\ell}_t^y\rangle\log_{|\mathcal{Y}|}\frac{1}{\langle\boldsymbol{w}_t,\boldsymbol{\ell}_t^y\rangle}}{|\mathcal{Y}|}\right)\right].$$

Let us assume the expected total loss of best policy is $\tilde{L}_{T,*}$. Thus, from Theorem 3, we get the expected cumulative loss

$$\mathbb{E}\left[R_T\right]=\mathbb{E}\left[\sum_{t=1}^{T}r_t\right]\le2|\mathcal{Y}|\sqrt{\frac{T\ln|\mathcal{Y}|\log|\Pi^*|}{\max\{\rho_T,\sqrt{1/T}\}}}+\tilde{L}_{T,*}.$$

Now plugging the regret bound $\mathcal{R}_T$ proved in Theorem 3 into the query complexity bound given by Lemma 12, we have

$$\sum_{t=1}^{T} \frac{\sum_{y \in \mathcal{Y}} \langle \boldsymbol{w}_t, \boldsymbol{\ell}_t^y \rangle \log_{|\mathcal{Y}|} \frac{1}{\langle \boldsymbol{w}_t, \boldsymbol{\ell}_t^y \rangle}}{|\mathcal{Y}|} \leq \frac{\left(2|\mathcal{Y}|\sqrt{\frac{T \ln |\mathcal{Y}| \log |\Pi^*|}{\max\{\rho_T, \sqrt{1/T}\}}} + \tilde{L}_{T,*}\right)\left(\log_{|\mathcal{Y}|} \frac{T^2(|\mathcal{Y}|-1)}{\left(2|\mathcal{Y}|\sqrt{\frac{T \ln |\mathcal{Y}| \log |\Pi^*|}{\max\{\rho_T, \sqrt{1/T}\}}}+\tilde{L}_{T,*}\right)^2}\right)}{|\mathcal{Y}|}$$

$$\leq \frac{\left(2|\mathcal{Y}|\sqrt{\frac{T \ln |\mathcal{Y}| \log |\Pi^*|}{\max\{\rho_T, \sqrt{1/T}\}}} + \tilde{L}_{T,*}\right)\left(\log_{|\mathcal{Y}|} T|\mathcal{Y}|\right)}{|\mathcal{Y}|}$$

$$= \frac{\left(2|\mathcal{Y}|\sqrt{\frac{T \ln |\mathcal{Y}| \log |\Pi^*|}{\max\{\rho_T, \sqrt{1/T}\}}} + \tilde{L}_{T,*}\right)\left(\log_{|\mathcal{Y}|} T + 1\right)}{|\mathcal{Y}|}.$$

Finally, by applying query complexity upper bound of Lemma 11, we got

$$\mathbb{E}\left[\sum_{t=1}^{T} U_t\right] \leq 2\sqrt{T} + \frac{\left(2|\mathcal{Y}|\sqrt{\frac{T \ln |\mathcal{Y}| \log |\Pi^*|}{\max\{\rho_T, \sqrt{1/T}\}}} + \tilde{L}_{T,*}\right)\left(\log_{|\mathcal{Y}|} T + 1\right)}{|\mathcal{Y}|}.$$

Since the second term on the RHS dominates the upper bound, we have

$$O\left(\mathbb{E}\left[\sum_{t=1}^{T} U_t\right]\right) = O\left(\frac{\left(\sqrt{\frac{T \log |\Pi^*|}{\max\{\rho_T, \sqrt{1/T}\}}} + \tilde{L}_{T,*}\right)(\ln T)}{\sqrt{\ln(|\mathcal{Y}|)}}\right) \overset{(a)}{=} O\left(\left(\sqrt{\frac{T \log |\Pi^*|}{\max\{\rho_T, \sqrt{1/T}\}}} + \tilde{L}_{T,*}\right)(\ln T)\right),$$

where step (a) is obtained by suppressing constant coefficients involving $|\mathcal{Y}|$ into the $O$ notation. $\quad\square$

# G  Additional Experiments

In this section, we further evaluate CAMS and provide additional experimental results (complementary to the main results presented in Fig. 2) under the following scenarios:

1. In App. G.1, we demonstrate that CAMS outperforms the baselines on a large scale dataset as well.

2. In App. G.2, we perform ablation study of three query strategies CAMS (entropy), variance and random strategy. CAMS *(Entropy)* achieves the minimum cumulative loss for CIFAR10, DRIFT, and VERTEBRAL under the same query cost and outperform the other query strategies.

3. In a *mixture of experts* environment, CAMS converges to the best policy and outperforms all others (App. G.3);

4. In a *non-contextual* (no experts) environment, CAMS has approximately equal performance as Model Picker to reach the best classifier effectively (App. G.4);

5. In an *adversarial* environment, CAMS can efficiently recover from the adversary and approach the performance of the best classifier (App. G.5);

6. In a *complete sub-optimal expert* environment, a variant of the CAMS algorithm, namely CAMS-MAX, which deterministically picks the most probable policy and selects the most probable model, outperforms CAMS-Random-Policy, which randomly samples a policy and selects the most probable model (App. G.7 & App. G.8). However, CAMS-MAX at most approaches the performance of the best policy. In contrast, perhaps surprisingly, CAMS is able to outperform the best policy on both VERTEBRAL and HIV (App. G.6).

7. In App. G.9, we summarize the maximum query cost under a fixed number of realizations with its associated cumulative loss for all baselines (exclude oracle) on all benchmarks in experiment section.

8. In App. G.10, we compare the query complexity of each baselines and demonstrate that CAMS has the lowest query cost increasing rate on CIFAR10, DRIFT and VERTEBRAL dataset.

9. In previous studies, we assume that the data comes in an online format. In App. G.11, we assume we know the data stream length ahead and applying the scaling parameter to each algorithm to query their top data points from hindsight. CAMS still outperforms all the baselines.

10. In App. G.12, we compare CAMS with CAMS-nonactive, a greedy version (query label for each incoming data point). Although CAMS query much less data, it still performs equally well or even better than the greedy version.

11. In App. G.13, we demonstrates that CAMS can achieve negative RCL on all benchmarks, which means it outperforms any algorithms that chase the best classifier where the horizontal 0 line represents the performance benchmark of best classifier.

## G.1  Performance of CAMS at scale: Experimental results on CovType

We scaled up our experiments on a larger dataset, CovType [24]. The CovType dataset offers details about different types of forest cover in the United States. It contains details including slope, aspect, elevation, measurements of the wilderness area, and the type of forest cover. CovType has 580K samples, of which 100K instances were chosen at random as online stream for testing. Fig. 4 demonstrated that CAMS outperforms all baselines which is consistent with the existing results in experiment section.

## G.2  Query strategies ablation comparison

Using the same CAMS model recommendation section, we compare three query strategies: the adaptive model-disagreement-based query strategy in Line 10-14 of Fig. 1 (referred to as *entropy* in the following), the variance-based query strategy from Model Picker [39] (referred to as *variance*), and a random query strategy. Fig. 5 shows that CAMS's adaptive query strategy has the sharpest converge rate on cumulative loss, which demonstrates the effectiveness of the queried labels. Moreover, *entropy* achieves the minimum cumulative loss for CIFAR10, DRIFT, and VERTEBRAL under the same query cost. For the HIV dataset, there is no clear winner between *entropy* and *variance* since the mean of their performance lie within the error bar of each other for the most part.

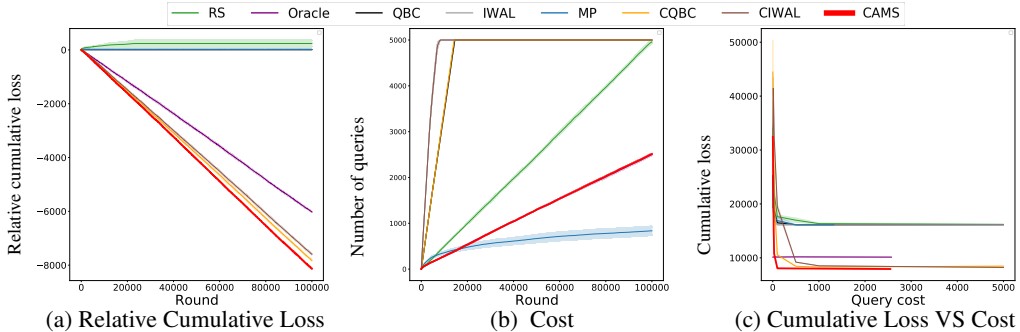

(a) Relative Cumulative Loss    (b) Cost    (c) Cumulative Loss VS Cost

Figure 4: Comparing CAMS with 7 baselines on CovType in terms of relative cumulative loss, query complexity, and cost effectiveness. CAMS outperforms all baselines. **(Left)** Performance measured by relative cumulative loss (i.e. loss against the best classifier) under a fixed query cost $B$ (where $B = 1000$). **(Middle)** Number of queries and **(Right)** Performance of cumulative loss by increasing the query cost, for a fixed number of rounds $T$ (where $T = 100,000$) and maximal query cost $B$ (where $B = 5000$). **Algorithms**: 4 contextual {Oracle, CQBC, CIWAL, CAMS} and 4 non-contextual baselines {RS, QBC, IWAL, MP} are included (see Section ). 90% confident interval are indicated in shades.

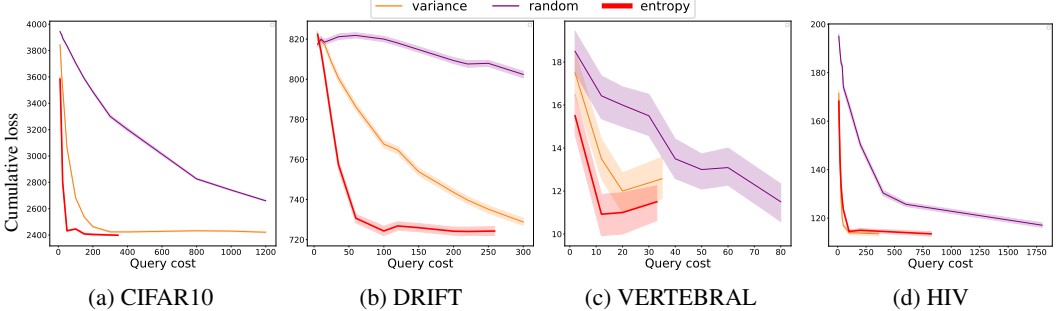

(a) CIFAR10    (b) DRIFT    (c) VERTEBRAL    (d) HIV

Figure 5: Ablation study of three query strategies (entropy, variance, random) for 4 diverse benchmarks based on the same model recommendation strategy. Under the same query cost constraint, CAMS's entropy-based strategy exceeds the performance of the other two strategies on non-binary benchmarks in terms of query cost and cumulative lost. 90% confident intervals are indicated in shades.

### G.3 Comparing CAMS with each individual expert

We evaluate CAMS by comparing it with all the policies available in various benchmarks. The policies in each benchmark are summarized in App. D.2 and Table D.5. The empirical results in Fig. 6 demonstrate that CAMS could efficiently outperform all policies and converge to the performance of the best policy with only slight increase in query cost in all benchmarks. In particular, on the VERTEBRAL and HIV benchmarks, CAMS even outperforms the best policy.

### G.4 Comparing CAMS against Model Picker in a context-free environment

CAMS outperforms Model Picker in Fig. 2, by leveraging the context information for adaptive model selection. In a context-free environment, $\Pi = \{\varnothing\}$, so $\Pi^* := \{\pi_1^{\mathrm{const}}, \ldots, \pi_k^{\mathrm{const}}\}$, where $\pi_j^{\mathrm{const}}(\cdot) := \boldsymbol{e}_j$ represents a policy that only recommends a fixed model. In this case, selecting the best policy to CAMS equals selecting the best single model. Fig. 7 demonstrates that the mean of CAMS and Model Picker lies in the shades of each other, which means CAMS has approximately the same performance as model picker considering the randomness on all benchmarks.

### G.5 Robustness against malicious experts in adversarial environments

When given only malicious and random advice policies, the conventional contextual online learning from experts advice framework will be trapped in the malicious or random advice. In contrast, CAMS could efficiently identify these policies and avoid taking advice from them. Meanwhile, it also successfully identifies the best classifier to learn to reach its best performance.

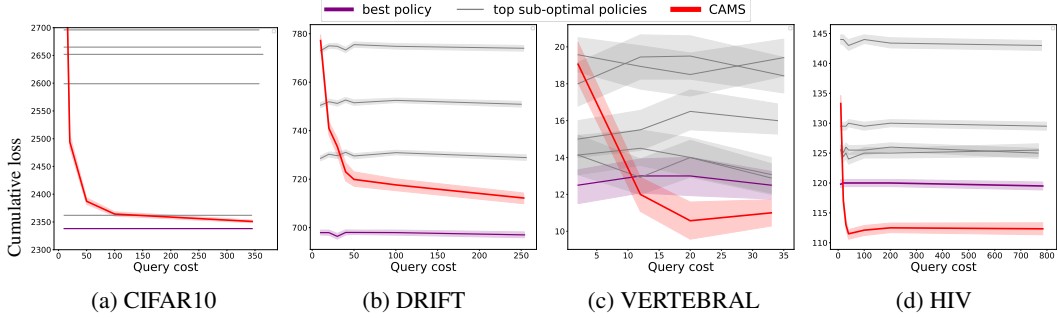

(a) CIFAR10      (b) DRIFT      (c) VERTEBRAL      (d) HIV

Figure 6: Comparing CAMS with every single policy (only plotted top performance policies in Figure). CAMS could approach the best expert and exceed all others with limited queries. In particular, on VERTEBRAL and HIV Benchmarks, CAMS outperforms the best expert. 90% confident intervals are indicated in shades.

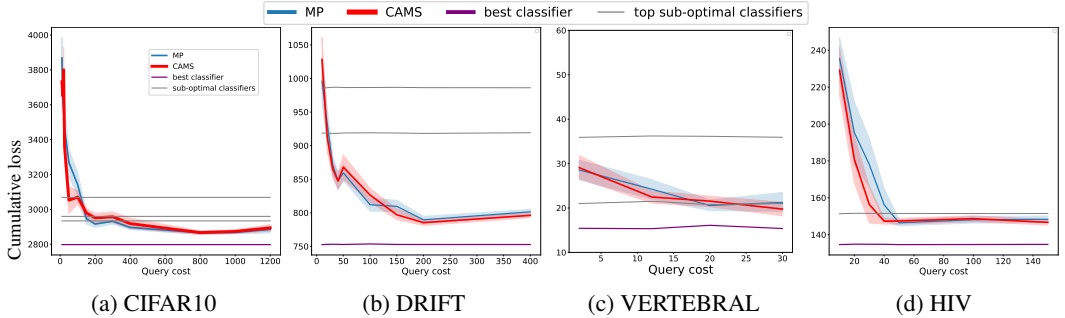

(a) CIFAR10      (b) DRIFT      (c) VERTEBRAL      (d) HIV

Figure 7: Comparing the model selection strategy of CAMS and Model Picker baseline based on the same variance-based query strategy in a context-free environment. CAMS has approximately the same performance as Model Picker on all the benchmarks. 90% confident intervals are indicated in shades.

The *novelty* in CAMS that enables this robustness is that we add the constant policies $\{\pi_1^{const}, \ldots, \pi_k^{const}\}$ into the policy set $\Pi$ to form the new set as $\Pi^*$. To illustrate the performance difference, we have created a variant of CAMS by adapting to the conventional approach (named CAMS-conventional). Fig. 8 demonstrates that CAMS could outperform all the malicious and random policies and converge to the performance of the best classifier. **CAMS-*conventional:*** We create the CAMS-conventional algorithm as the CAMS using policy set $\Pi$, not $\Pi^*$.

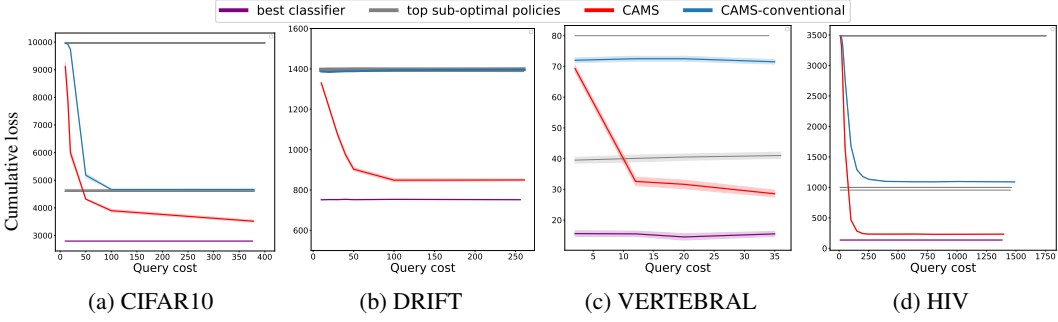

(a) CIFAR10      (b) DRIFT      (c) VERTEBRAL      (d) HIV

Figure 8: Evaluating the robustness of CAMS compared to the conventional learning from experts' advice (CAMS-conventional) in a complete malicious and random policies environment. When no good policy is available, CAMS could recover from malicious advice and successfully approach the performance of the best classifier. In contrast, the conventional approach will be trapped in malicious advice. 90% confident intervals are indicated in shades.

### G.6 Outperformance over the best policy/expert

We also observe that CAMS does not stop at approaching the best policy or classifier performance. Sometimes, it even outperforms all the policies and classifiers, and Fig. 9 demonstrates such a case. To demonstrate the advantage of CAMS, we create two variant versions of CAMS: (1) CAMS-MAX (App. G.7), (2) CAMS-Random-Policy (App. G.8). CAMS-MAX and CAMS-Random-Policy use the same algorithm as CAMS in adversarial settings but have different model selection strategies for ablation study in the stochastic settings.

We evaluate the three algorithms on VERTEBRAL and HIV benchmarks in terms of (a) *normal policies* (Fig. 9 Left), (b) *classifiers* (Fig. 9 Middle), and (c) *malicious and random policies* (Fig. 9 Right). In the normal policies column, we only compare the policies with regular policies giving helpful advice. In the classifier column, we compare them with the performance of classifiers only. In the malicious and random policies column, we compare them with unreasonable policies only.

Fig. 9 demonstrates that all three algorithms could outperform the malicious/random policies. However, CAMS-Random-Policy does not outperform the best classifier while both CAMS and CAMS-MAX can on both benchmarks. CAMS-MAX approaches the performance of the best policy but does not outperform the best policy on both benchmarks. Finally, perhaps surprisingly, CAMS outperforms the best policy (Oracle) on both benchmarks and continues to approach the hypothetical, optimal policy (with 0 cumulative loss).

This surprising factor is contributed by the adaptive weighted policy of CAMS, which adaptively creates a better policy by combining the advantage of each sub-optimal policy and classifier to reach the performance of the hypothetical, optimal policy (defined as $\sum_{t=1}^{T} \min_{i \in [n+k]} \widetilde{\ell}_{t,i}$). The second reason could be that the benchmark we created, or any real-world cases, will not be strictly in a stochastic setting (in which a single policy outperforms all others or has lower $\mu$ in every round). The weight policy strategy can make a better combination of advice for this case.

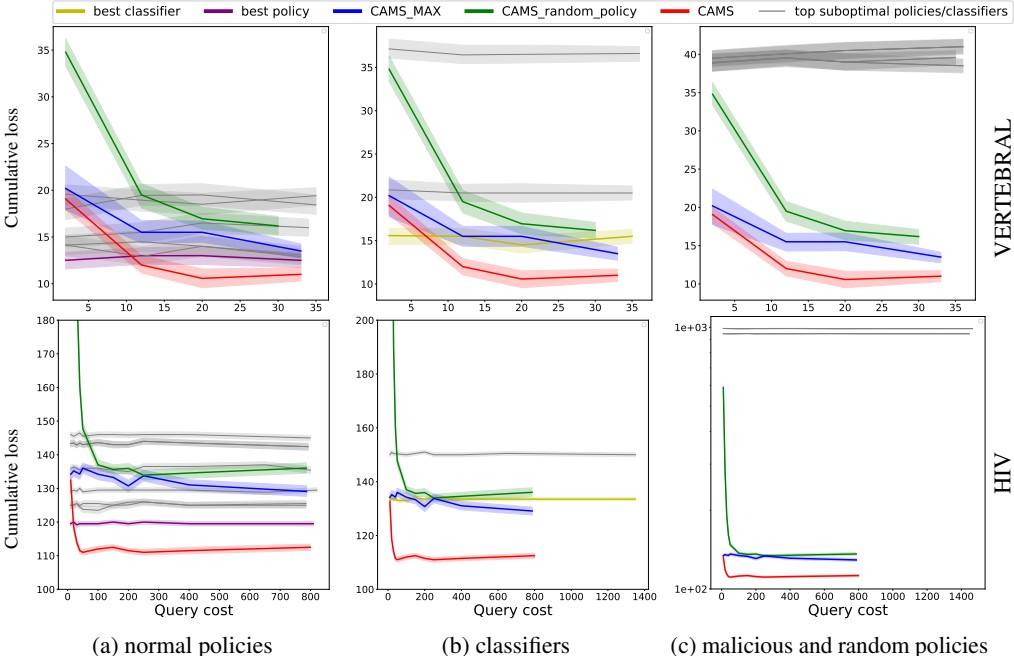

(a) normal policies        (b) classifiers        (c) malicious and random policies

Figure 9: Comparing CAMS, CAMS-MAX and CAMS-RANDOM-POLICY with top policies and classifiers in the VERTEBRA and HIV benchmarks. They outperform all the malicious/random policies. Moreover, CAMS and CAMS-MAX outperform the best classifier. Finally, only CAMS even exceeds the best policy (Oracle) in both benchmarks and continues approaching the hypothetical, optimal policy (0 cumulative loss). 90% confident intervals are indicated in shades.

## G.7 The CAMS-MAX algorithm

CAMS-MAX is a variant of CAMS. In an adversarial setting, they share the same algorithm. However, in a stochastic setting, CAMS-MAX gets the index $i^*$ of max value in the probability distribution of policy $\boldsymbol{q}$, and selects the model with the max value in $\pi_{i^*}(\boldsymbol{x}_t)$ to recommendation. The difference is marked as blue color in Fig. 10.

```
1:  Input: Models F, policies Π*, #rounds T, budget b
2:  Initialize loss L̃₀ ← 0; query cost C₀ ← 0          21: procedure SETRATE(t, xₜ, m)
3:  for t = 1, 2, ..., T do                            22:     if STOCHASTIC then
4:      Receive xₜ                                      23:         ηₜ = √(ln m / t)
5:      ηₜ ← SETRATE(t, xₜ, |Π*|)                      24:     end if
6:      Set q_{t,i} ∝ exp(−ηₜL̃_{t−1,i}) ∀i ∈ |Π*|     25:     if ADVERSARIAL then
7:      jₜ ← RECOMMEND(xₜ, qₜ)                         26:         Set ρₜ as in adversarial section
8:      Output ŷ_{t,jₜ} ∼ f_{t,jₜ} as the prediction for xₜ  27:     ηₜ = √(1/√t + ρₜ/(c²ln c)) · √(ln m / T)
9:      Compute zₜ in Eq. (4)                          28:     end if
10:     Sample Uₜ ∼ Ber(zₜ)                            29:     return ηₜ
11:     if Uₜ = 1 and Cₜ ≤ b then                      30: end procedure
12:         Query the label yₜ
13:         Cₜ ← C_{t−1} + 1
14:         Compute ℓₜ: ℓ_{t,j} = 𝕀{ŷ_{t,j} ≠ yₜ}, ∀j ∈ [|F|]   29: procedure RECOMMEND(xₜ, qₜ)
15:         Estimate model loss: ℓ̂_{t,j} = ℓ_{t,j}/zₜ, ∀j ∈ [|F|]  30:     if STOCHASTIC then
16:         ℓ̃ₜ: ℓ̃_{t,i} ← ⟨πᵢ(xₜ), ℓ̂_{t,j}⟩, ∀i ∈ [|Π*|]  31:         iₜ ← maxind(qₜ)
17:         L̃ₜ = L̃_{t−1} + ℓ̃ₜ                        32:         jₜ ← maxind(π_{iₜ}(xₜ))
18:     else                                           33:     end if
19:         L̃ₜ = L̃_{t−1}                               34:     if ADVERSARIAL then
20:         Cₜ ← C_{t−1}                                35:         iₜ ∼ qₜ
21:     end if                                         36:         jₜ ∼ π_{iₜ}(xₜ)
22: end for                                            37:     end if
                                                       38:     return jₜ
                                                       39: end procedure
```

Figure 10: The CAMS-MAX Algorithm

## G.8 The CAMS-Random-Policy algorithm

```
1:  Input: Models F, policies Π*, #rounds T, budget b
2:  Initialize loss L̃₀ ← 0; query cost C₀ ← 0          21: procedure SETRATE(t, xₜ, m)
3:  for t = 1, 2, ..., T do                            22:     if STOCHASTIC then
4:      Receive xₜ                                      23:         ηₜ = √(ln m / t)
5:      ηₜ ← SETRATE(t, xₜ, |Π*|)                      24:     end if
6:      Set q_{t,i} ∝ exp(−ηₜL̃_{t−1,i}) ∀i ∈ |Π*|     25:     if ADVERSARIAL then
7:      jₜ ← RECOMMEND(xₜ, qₜ)                         26:         Set ρₜ as in adversarial section
8:      Output ŷ_{t,jₜ} ∼ f_{t,jₜ} as the prediction for xₜ  27:     ηₜ = √(1/√t + ρₜ/(c²ln c)) · √(ln m / T)
9:      Compute zₜ in Eq. (4)                          28:     end if
10:     Sample Uₜ ∼ Ber(zₜ)                            29:     return ηₜ
11:     if Uₜ = 1 and Cₜ ≤ b then                      30: end procedure
12:         Query the label yₜ
13:         Cₜ ← C_{t−1} + 1
14:         Compute ℓₜ: ℓ_{t,j} = 𝕀{ŷ_{t,j} ≠ yₜ}, ∀j ∈ [|F|]   29: procedure RECOMMEND(xₜ, qₜ)
15:         Estimate model loss: ℓ̂_{t,j} = ℓ_{t,j}/zₜ, ∀j ∈ [|F|]  30:     if STOCHASTIC then
16:         ℓ̃ₜ: ℓ̃_{t,i} ← ⟨πᵢ(xₜ), ℓ̂_{t,j}⟩, ∀i ∈ [|Π*|]  31:         iₜ ∼ qₜ
17:         L̃ₜ = L̃_{t−1} + ℓ̃ₜ                        32:         jₜ ← maxind(π_{iₜ}(xₜ))
18:     else                                           33:     end if
19:         L̃ₜ = L̃_{t−1}                               34:     if ADVERSARIAL then
20:         Cₜ ← C_{t−1}                                35:         iₜ ∼ qₜ
21:     end if                                         36:         jₜ ∼ π_{iₜ}(xₜ)
22: end for                                            37:     end if
                                                       38:     return jₜ
                                                       39: end procedure
```

Figure 11: The CAMS-Random-Policy Algorithm

CAMS-Random-Policy is a variant of CAMS. It shares the same algorithm with CAMS in an adversarial environment. However, it uses a random sampling policy method in a stochastic setting.

It randomly samples the policy from the probability distribution of policy $q$, and selects the model with max value in $\pi_{i^*}(x_t)$ to recommendation. The difference is marked as blue color in Fig. 11.

## G.9 Maximal queries from experiments

Table 6 in this section summarizes the maximum query cost for a given data stream (of fixed total size), with its associated cumulative loss for all baselines (exclude Oracle) on all benchmarks in experiment section. The result in this table is slightly different from the query complexity curves of Fig. 2 (Middle). The curve in Fig. 2 (Middle) takes the average value, while the table takes the maximal value from a fixed number of simulations. CAMS wins over all baselines (other than Oracle) in terms of query cost on CIFAR10, DRIFT, and VERTEBRAL benchmarks. CAMS outperforms all baselines in terms of cumulative loss on DRIFT, VERTEBRAL, and HIV benchmarks. In particular, CAMS outperforms both cumulative loss and query cost on the DRIFT and VERTEBRAL benchmarks.

| Algorithm | CIFAR10 | DRIFT | VERTEBRAL | HIV |
|---|---|---|---|---|
| *Max queries, Cumulative loss* | *1200, 10000* | *2000, 3000* | *80, 80* | *2000, 4000* |
| RS | 1200, 2916 | 2000, 766 | 80, 19 | 2000, 143 |
| QBC | 1200, 2857 | 1904, 771 | 72, 20 | 2000, 139 |
| IWAL | 1200, 2854 | 2000, 760 | 80, 19 | 690, 140 |
| MP | 1200, 2885 | 493, 803 | 33, 25 | 153, 148 |
| CQBC | 1200, 2284 | 1900, 744 | 68, 13 | 2000, 124 |
| CIWAL | 1200, 2316 | 2000, 746 | 80, 12 | 690, 124 |
| **CAMS** | **348**, 2348 | **251, 710** | **32, 11** | 782, **112** |

Table 5: Maximal queries from experiments

## G.10 Query complexity

To achieve the same level of prediction accuracy (measured by average cumulative loss over a fixed number of rounds), CAMS incurs less than 10% of the label cost of the best competing baselines on CIFAR10 (10K examples), and 68% the cost on VERTEBRAL (see Fig. 12); Fig. 12 [8] and Table 6 also demonstrate the compelling effectiveness of CAMS's query strategy outperforming all baselines in terms of query cost in VERTEBRAL, DRIFT, and CIFAR10 benchmarks, which is consistent with our query complexity bound in Theorem 2.

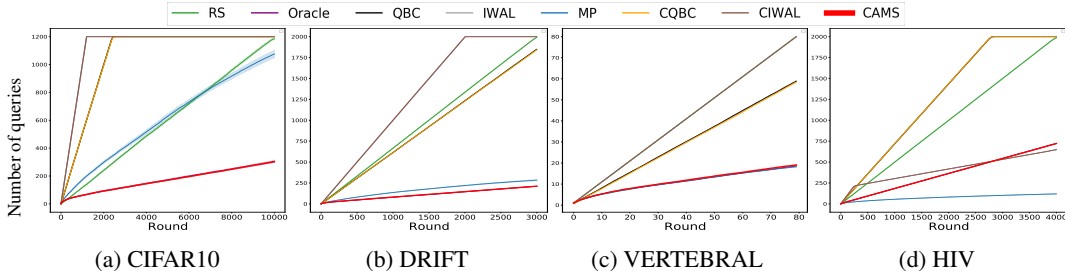

(a) CIFAR10    (b) DRIFT    (c) VERTEBRAL    (d) HIV

Figure 12: Comparing CAMS with 7 baselines on 4 diverse benchmarks in terms of query complexity (Number of queries). CAMS outperforms all baselines for a fixed number of rounds $T$ (where $T = 10000, 3000, 80, 4000$ from left to right) and maximal query cost $B$ (where $B = 1200, 2000, 80, 2000$ from left to right). **Algorithms**: 4 contextual {Oracle, CQBC, CIWAL, CAMS} and 4 non-contextual baselines {RS, QBC, IWAL, MP} are included (see Section ). 90% confident interval are indicated in shades.

## G.11 Fine-tuning the query probabilities for stochastic streams

For the experimental results we reported in the main paper, we consider a streaming setting where the data arrives online in an *arbitrary order* and *arbitrary length*. Therefore, for both CAMS and

---

[8]We also consider variants for each algorithm (other than Random and Oracle) where we scale the query probabilities based on the early-phase performance and observe similar behavior. See App. G.11 for the corresponding results.

the baselines, we used the exact off-the-shelf query criteria as described in experiment setup section *without fine-tuning the query probabilities*, which could be otherwise desirable in certain scenarios (e.g. for stochastic streams, where the query probability can be further optimized).

In this section, we consider such scenarios, and conduct an additional set of experiments to further demonstrate the performance of CAMS assuming stochastic data streams. Given the stream length $T$ and query budget $b$, we may optimize each algorithm by scaling their query probabilities, so that each algorithm allocates its query budget to the top $b$ informative labels in the entire online stream based on its own query criterion. Note that in practice, finding the exact scaling parameter is infeasible, as we do not know the online performance unless we observe the entire data stream. While it is challenging to determine the scaling factor for each algorithm under the adversarial setting, one can effectively estimate the scaling factor for stochastic streams, where the context arrives *i.i.d.*.

Concretely, we use the early budget to decide the scaling parameter in our following evaluation: Firstly, we use a small fraction (i.e. $T/10$) of the online stream and see how much queries $b_{\text{early}}$ each algorithm consumed. Then we calculate the scaling parameter $s = \frac{(b-b_{\text{early}})}{T-T/10} \cdot \frac{T/10}{b_{\text{early}}}$ and multiply the scaling factor with the query probability of each algorithm for the remaining $\frac{9}{10} \cdot T$ rounds. The results in Fig. 13 demonstrate that CAMS still outperforms all the baselines (excluding Oracle) when all algorithms select the top $b$ data of the whole online stream to query. The improvement of CAMS over the baseline approaches *does not differ much* between the two versions (with or without scaling) of the experiments as shown in the bottom plots of Fig. 2 and Fig. 13.

For a head-to-head comparison between the bottom plots of Fig. 2 and Fig. 13, note that the total number of rounds stays the same for DRIFT ($T = 3000$), VERTEBRAL ($T = 80$), and HIV ($T = 4000$); while we used half the rounds and half the maximal budget for CIFAR10 ($T = 5000$) for the version with scaling. Roughly speaking, the cumulative regret plots for the baselines were "strched out" to cover the full allocated budget after scaling, but we do not observe a significant difference in terms of the absolute gain in terms of the cumulative loss. Another way to read the difference between the two plots is to compare the cumulative losses at the budget range where all algorithms were not cut off early: e.g., for DRIFT, when Query Cost is 250, the cumulative losses for the competing algorithm stay roughly the same under the two evaluation scenarios.

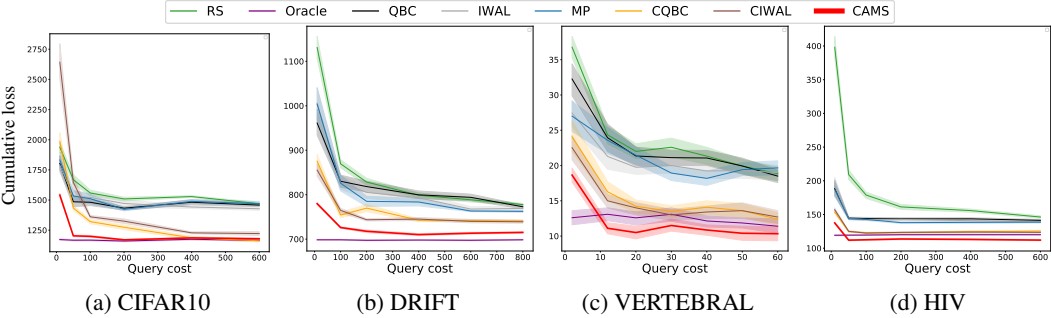

(a) CIFAR10      (b) DRIFT      (c) VERTEBRAL      (d) HIV

Figure 13: Comparing CAMS with 7 model selection baselines on 4 diverse benchmarks in terms of cost effectiveness after applying the scaling parameter to each algorithm. CAMS outperforms all baselines (excluding Oracle). Performance of cumulative loss by increasing the query cost, for a fixed number of rounds $T$ (where $T = 5000, 3000, 80, 4000$ from left to right) and maximal query cost $B$ (where $B = 600, 800, 60, 600$ from left to right). **Algorithms**: 4 contextual {Oracle, CQBC, CIWAL, CAMS} and 4 non-contextual baselines {RS, QBC, IWAL, MP} are included (see Section ). 90% confident interval are indicated in shades.

## G.12    Ablation study on the active query strategy

In this section, we compare the performance of CAMS and its non-active variant (CAMS-nonactive), which queries the label for each incoming data point. As shown in Fig. 14, CAMS performs equally well or better than CAMS-nonactive, even though it queries significantly less data. Surprisingly, on the DRIFT dataset, CAMS significantly outperforms CAMS-nonactive, even when using less than 10 percent of the query budget (Fig. 14b). This demonstrates that CAMS selectively choose the data to query to maximal optimize policy improvement, while CAMS queries all data points, regardless of their usefulness or noise, which hampers policy improvement and convergence.

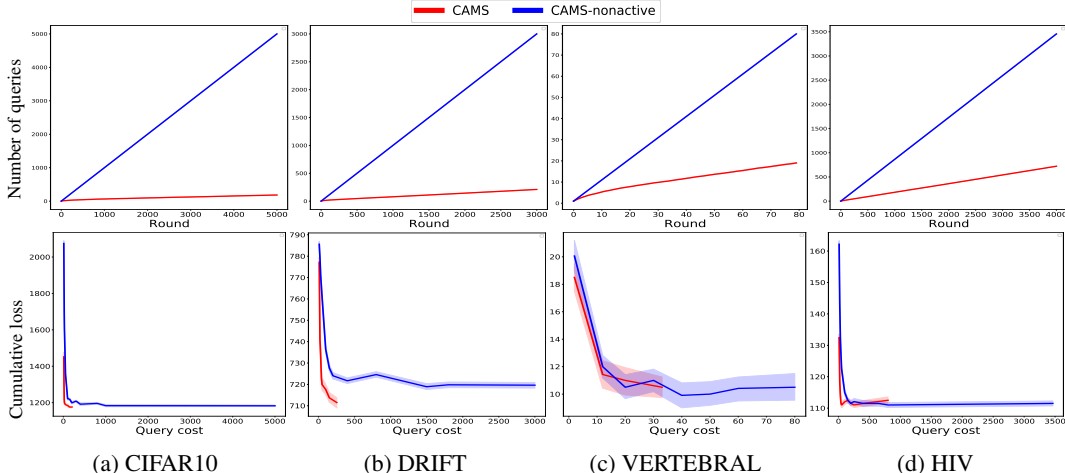

Figure 14: Comparing CAMS (in red) with CAMS-nonactive (in blue) on 4 diverse benchmarks in terms of query complexity, and cost effectiveness. CAMS outperforms or performs equally well to CAMS-nonactive with much less queried labels for all benchmarks. **(Top)** Number of queries and **(Bottom)** Performance of cumulative loss by increasing the query cost, for a fixed number of rounds $T$ (where $T = 5000, 3000, 80, 4000$ from left to right) and maximal query cost $B$ (where $B = T = 5000, 3000, 80, 4000$ from left to right). 90% confident interval are indicated in shades.

### G.13 Relative Cumulative Loss

**Relative cumulative loss (RCL).** At round $t$, we define RCL as $L_{t,j_i} - L_{t,j^*}$, where $L_{t,j^*}$ stands for the cumulative loss (CL) of the policy always selecting the best classifier, and $L_{t,j_i}$ stands for the CL of policy $i$.

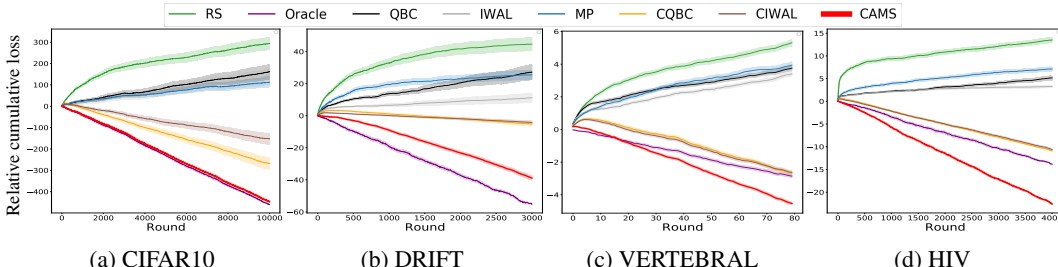

Figure 15: Comparing CAMS with 7 baselines on 4 diverse benchmarks in terms of loss trajectory. CAMS outperforms all baselines. Performance measured by relative cumulative loss (i.e. loss against the best classifier) under a fixed query cost $B$ (where $B = 200, 400, 30, 400$ from left to right). **Algorithms**: 4 contextual {Oracle, CQBC, CIWAL, CAMS} and 4 non-contextual baselines {RS, QBC, IWAL, MP} are included (see Section ). 90% confident interval are indicated in shades.

The RCL under the same query cost for all baselines is shown in Fig. 15. The loss trajectory demonstrates that CAMS efficiently adapts to the best policy after only a few rounds and outperforms all baselines in all benchmarks. The result also demonstrates that CAMS can achieve negative RCL on all benchmarks, which means it outperforms any algorithms that chase the best classifier, as the horizontal 0 line represents the performance benchmark of best classifier. This empirical result aligns with Theorem 1 that, in the worst scenario, if the best classifier is the best policy, CAMS will achieve its performance. Otherwise, CAMS will reach a better policy and incurs no regret.

CAMS could achieve such performance because when an Oracle fails to achieve 0 loss over all instances and contexts, CAMS has the opportunity to outperform the Oracle *in those rounds Oracle does not make the best recommendation*. For instance, the stochastic version of CAMS (Line 22-23; Line 30-32 in Fig. 1) may achieve this by recommending a model using the weighted majority vote among all policies. Therefore, one can view CAMS as adaptively constructing a new policy at each

round by combining the advantages of each sub-optimal policy, which may outperform any single expert/policy. Furthermore, for the experiments we ran (or in most real-world scenarios), the data streams are not strictly in a stochastic setting (in which a single policy outperforms all others or has a lower expected loss in every round). The weighted policy strategy may find a better combination of "advices" in such cases (see Fig. 2 and App. G.6).

## Global Rebuttal Response

## H    Experiments on ImageNet

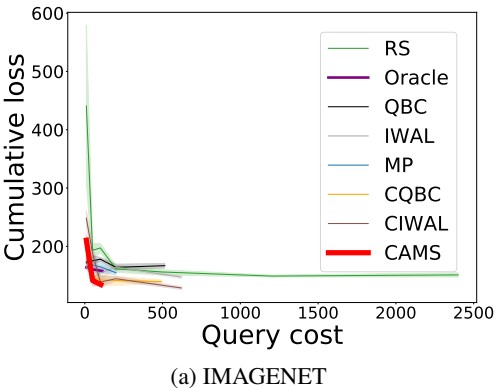

(a) IMAGENET

Figure 16: Comparison of CAMS with 7 baselines on IMAGENET benchmark in terms of cost effectiveness. We plot the cumulative loss as we increase the query cost for a fixed number of rounds $T$ and maximal query cost $B$ ($T = 3000$, and $B = 2500$). CAMS outperforms all baselines. **Algorithms**: 4 contextual {Oracle, CQBC, CIWAL, CAMS} and 4 non-contextual baselines {RS, QBC, IWAL, MP} are included. 90% confident interval are indicated in shades.

## I    Comparing CAMS against recent works in active learning

| Active Learning Setting / Algorithms | Coreset | Batch-BALD | BADGE; VAAL; ClusterMargin | BALANCE; GLISTER | VeSSAL | Model Picker | CAMS |
|---|---|---|---|---|---|---|---|
| *Streaming, sequential* | × | × | × | × | × | ✓ | ✓ |
| *Streaming, batch* | × | × | × | × | ✓ | × | × |
| *Pool-based, batch* | ✓ | ✓ | ✓ | ✓ | × | × | × |

Table 6: Selective comparison against recent works in active learning. Among these algorithms, Coreset (Sener & Savarese, 2017) is a diversity sampling strategy for deep active learning; Batch-BALD is an uncertainty sampling strategy; BADGE (Ash et al., 2019), VAAL (Sinha et al., 2019) ClusterMargin (Citovsky et al., 2021), and VeSSAL (Saran et al., 2023) represent strategies that combine both; GLISTER (Killamsetty et al., 2020) and BALANCE (Zhang et al., 2023) represent decision-theoretic approaches that directly optimize the utility of queries.

