# OpenReview forum: "Contextual Active Model Selection"
_NeurIPS.cc/2024/Conference — NeurIPS 2024 poster_

### Official Review · Reviewer_E9VN · 2024-07-09

**Soundness:** 2
**Presentation:** 3
**Contribution:** 2
**Rating:** 4
**Confidence:** 3

**Summary:**

This paper focuses on the online contextual active model selection problem. Specifically, the learner receives an unlabeled data point as a context at each round, and the objective is to adaptively select the best model to predict while limiting label requests. To address this problem, the authors proposed a CAMS method that contains a contextual active model selection algorithm and an active query component. Theoretical results about regret in both adversarial and stochastic settings are provided.

**Strengths:**

1) The problem setting that we need to do a model selection at each around is novel and interesting.

2) Theoretical analysis on both adversarial and stochastic settings is provided.

**Weaknesses:**

About the problem setting, I have some concerns and questions:

1) In different tasks, such as image classification, and tabular data, we may have many different pre-trained models. For example, in the image classification tasks, we may choose the deep neural network trained on the ImageNet, or we can also adopt the CLIP model. How to construct the candidate model pool?

2) Given a model pool, in the first round, we have already selected a model. How do we decide whether we need to choose a new model or continue using the previous one?

3) In the proposal, given a new instance, the algorithm needs to run each candidate model on the instance, which requires a lot of computation cost. There are also some methods proposed to assign each model a specification to describe its functionality[1]. Can these methods be combined with the proposal?

4) In many cases, we may need to ensemble multiple models, can the proposal be extended to multiple model selection?

[1] Lan-Zhe Guo, Zhi Zhou, Yu-Feng Li, Zhi-Hua Zhou. Identifying Useful Learnwares for Heterogeneous Label Spaces. In: Proceedings of the 40th International Conference on Machine Learning (ICML 2023), Hawaii, 2023. Page: 12122-12131.

**Questions:**

As discussed above.

---

> ### Author Rebuttal · Authors · 2024-08-07
>
> Thank you for the feedback on our work! Below please find our detailed responses to your questions.
>
> ---
> > ***Q1:*** "In different tasks, such as image classification, and tabular data, we may have many different pre-trained models. For example, in the image classification tasks, we may choose the deep neural network trained on the ImageNet, or we can also adopt the CLIP model. How to construct the candidate model pool?"
>
> ***A:*** Thanks for letting us know your confusion. Ideally, it is beneficial to have diverse models, such as pretrained models with different structures or access to varied data distributions during the pre-training stage. CAMS learns to leverage the unique strengths of these black-box models through an online process, actively querying labels and policies. This approach tailors the queries to different contexts to distinguish the capabilities of the various policies and models.
> To further address your concerns, we conducted additional experiments using a more recent and complex large-scale dataset, the ImageNet dataset with 1000 categories. We also incorporated six more recent pretrained models, including CLIP, Inception V4, VGG19, and PNASNet. In Fig.16 of the attached global response PDF, we present our studies on cost-effective query experiments with the ImageNet dataset using these newer pretrained models. The results are consistent with our previous findings, demonstrating that CAMS outperforms all baselines. CAMS not only shows significant superiority over both contextual and non-contextual baselines but also achieves the lowest label query cost compared to existing baselines and the current state-of-the-art, ModelPicker [1].
>
> [1] Online Active Model Selection for Pre-trained Classifiers, AISTATS 2021
>
> > ***Q2:*** "Given a model pool, in the first round, we have already selected a model. How do we decide whether we need to choose a new model or continue using the previous one?"
>
> ***A:*** Thank you for the question. This selection process will be determined by the CAMS algorithm, which is quite dynamic. In each round, CAMS will select a model by considering policy advice regarding the model's performance in the current context. It will choose the model with the highest probability by leveraging advice from multiple experts in a stochastic setting. In an adversarial setting, it will first sample a policy based on the policies' exponential cumulative loss and then sample the model according to the policy's advice distribution.
>
>
> > ***Q3:*** "In the proposal, given a new instance, the algorithm needs to run each candidate model on the instance, which requires a lot of computation cost. There are also some methods proposed to assign each model a specification to describe its functionality[1]. Can these methods be combined with the proposal?"
>
> ***A:*** Thanks for raising the question. Our work is quite different from the Learnware [1] setting. The main differences include the following:
> * We treat models and policies as black boxes, whereas Learnware requires assigning a specification SSS to each model, which involves parameters of the reduced model.
> * CAMS focuses on active querying and cost-effectiveness, which Learnware does not.
> * We assume that these pretrained black-box models are for the same task but with different expertise, aiming to combine their complementary expertise. In contrast, Learnware assumes models are for different tasks.
>
> Therefore, they operate in quite different settings, making it challenging to combine these methods.
>
> [1] Identifying Useful Learnwares for Heterogeneous Label Spaces. ICML 2023
>
>
> > ***Q4:*** "In many cases, we may need to ensemble multiple models, can the proposal be extended to multiple model selection?"
>
> ***A:*** Thanks for the question. This approach can be adapted for multiple model selection scenarios by modifying the RECOMMEND part (Fig 1, lines 29-39) of the CAMS algorithm. Rather than selecting the top-ranking model for a given instance to query in a stochastic setting, we could simply select a few top candidate models to ensemble the models' predictions.
>
>
> ---
>
> We hope these responses adequately address your concerns. We appreciate your feedback and look forward to further discussions. Thank you!

---

> > ### Comment · Area_Chair_utKU · 2024-08-12
> > **Please update your review and engage with the authors**
> >
> > Dear reviewer,
> >
> > Please provide an update to your review. The authors have provided a quite substantial rebuttal. Please acknowledge that you have read the rebuttal, and please post any questions that you may still have. Also clarify if you want to adjust your score.
> >
> > Many thanks,
> > Your AC.

---

> ### Comment · Area_Chair_utKU · 2024-08-12
> **Please update your review and engage with the authors**
>
> Dear reviewer,
>
> Please provide an update to your review. The authors have provided a quite substantial rebuttal. Please acknowledge that you have read the rebuttal, and please post any questions that you may still have. Also clarify if you want to adjust your score.
>
> Many thanks, Your AC.
>
> PS: Sorry for the double message, but this is now a reply to the review, so I hope this will automatically reach your inbox now.

---

> ### Comment · Area_Chair_utKU · 2024-08-13
> **Please respond**
>
> Dear reviewer,
>
> Thanks again for your thoughtful review. As this paper is a bit borderline, I would really like to know if the rebuttal had any affect on your review.
>
> Therefore please provide an update to your review and acknowledge that you have read the rebuttal and clarify if you want to adjust your score.
>
> Many thanks, Your AC.

---

### Official Review · Reviewer_FX2o · 2024-07-12

**Soundness:** 3
**Presentation:** 2
**Contribution:** 2
**Rating:** 5
**Confidence:** 3

**Summary:**

The paper introduces CAMS, an algorithm designed for online contextual active model selection. CAMS minimizes labeling costs by selecting the most appropriate pre-trained models for given contexts and strategically querying labels. The paper provides theoretical analysis of regret and query complexity in both adversarial and stochastic settings. Empirical evaluations on benchmark tasks such as CIFAR10 and DRIFT show that CAMS reduces labeling effort significantly while maintaining or improving accuracy.

**Strengths:**

- The integration of a contextual model selection mechanism with an active query strategy is a novel approach that effectively addresses the challenge of selecting the best model for varying data contexts while minimizing labeling costs.
- The paper offers a robust theoretical foundation, with detailed proofs and analyses of regret and query complexity.
- The empirical results are strong, showing that CAMS significantly reduces labeling effort while maintaining or improving accuracy across various benchmarks, including CIFAR10 and DRIFT.

**Weaknesses:**

-  The method does not discuss how it handles dynamic updates to the set of classifiers or policies. In practical applications, the set of available models may change over time. The lack of a mechanism to incorporate such updates limits the robustness and adaptability of the proposed solution.
- While the datasets chosen provide a range of scenarios, the empirical evaluation could benefit from more recent and complex datasets to better demonstrate CAMS' capabilities. Additionally, the comparisons are primarily with older methods or basic variants, and a broader set of state-of-the-art methods, including recent advances in contextual bandits and active learning, are not fully explored.
- Some of the mathematical notations and their explanations are dense and could be clarified further. For instance, the derivation and intuition behind the exponential weighting and the specific choice of ηt could be more thoroughly explained to improve understanding.
- The paper does not discuss the stability of empirical results across multiple runs. Given that the evaluation relies on certain stochastic processes, it would be beneficial to report the variance or standard deviation of the results to provide a clearer picture of the method's reliability.

**Questions:**

- How does CAMS handle scenarios where the set of pre-trained classifiers is not fixed or continuously updated?
- Can more details be provided on the selection and construction of the policy set used in experiments?
- How does the method perform when applied to tasks beyond classification, such as regression or ranking problems?

**Limitations:**

The author mentions the limited focus on classification and non-uniform loss functions in this work.

---

> ### Author Rebuttal · Authors · 2024-08-07
>
> Thank you for your detailed review of our work! We greatly appreciate your recognition of the novelty of our approach, the robust and thorough theoretical foundation, and the strong empirical results of CAMS. Below, you will find our detailed responses to your questions.
>
> ---
>
> > ***Q1:*** "The method does not discuss how it handles dynamic updates to the set of classifiers or policies. In practical applications, the set of available models may change over time. The lack of a mechanism to incorporate such updates limits the robustness and adaptability of the proposed solution."
>
> ***A:*** Thanks for raising the question! Yes, this is an interesting setting. In this work, we only consider the scenario of having access to pretrained models. However, in our adversarial setting, the regret bound and query complexity could address your concern and provide performance guarantees for worst-case scenarios, including cases where the available models change over time.
>
> > ***Q2:*** "While the datasets chosen provide a range of scenarios, the empirical evaluation could benefit from more recent and complex datasets to better demonstrate CAMS' capabilities. Additionally, the comparisons are primarily with older methods or basic variants, and a broader set of state-of-the-art methods, including recent advances in contextual bandits and active learning, are not fully explored."
>
> ***A:*** Thanks for raising the question! To address your concerns, we conducted additional experiments using a more recent and complex large-scale dataset, the ImageNet dataset with 1000 categories. We also incorporated six more recent pretrained models, including CLIP, Inception V4, VGG19, and PNASNet. In Figure 16 of the attached global response PDF, we present our studies on cost-effective query experiments with the ImageNet dataset using these newer pretrained models. The results are consistent with our previous findings, demonstrating that CAMS outperforms all baselines. CAMS not only shows significant superiority over both contextual and non-contextual baselines but also achieves the lowest label query cost compared to existing baselines and the current state-of-the-art, ModelPicker [1].
>
> In addition, by comparing our approach with a broader set of state-of-the-art active learning methods, as shown in the following table, we illustrate that the most recent baseline with a setting closest to CAMS is ModelPicker [1].
>
>
>
> | **Active Learning Setting \ Algorithms** | **Coreset (2017)** | **Batch-BALD (2019)** | **BADGE (2019); VAAL (2019); ClusterMargin (2021)** | **BALANCE (2023); GLISTER (2020)** | **VeSSAL (2023)** | **Model Picker (2021)** | **CAMS** |
> |-------------------------------------------|-------------|----------------|-------------------------------|----------------------|------------|------------------|----------|
> | *Streaming, sequential*                   | ×           | ×              | ×                             | ×                    | ×          | ✔️               | ✔️       |
> | *Streaming, batch*                        | ×           | ×              | ×                             | ×                    | ✔️         | ×                | ×        |
> | *Pool-based, batch*                       | ✔️          | ✔️             | ✔️                            | ✔️                   | ×          | ×                | ×        |
>
>
> [1] Online Active Model Selection for Pre-trained Classifiers, AISTATS 2021
>
>
> > ***Q3:*** "Some of the mathematical notations and their explanations are dense and could be clarified further. For instance, the derivation and intuition behind the exponential weighting and the specific choice of ηt could be more thoroughly explained to improve understanding."
>
> ***A:*** The derivation and intuition behind exponential weighting is: 1.Bias towards Better Predictions and effective penalty for poor predictions: It increases the weight of consistently accurate experts, quickly focusing on the best performers and improving overall performance. It significantly reduces the weight of poor predictors, minimizing their influence and maintaining robust decision-making. 2. Balancing Exploration and Exploitation: It balances exploration and exploitation by occasionally giving higher weights to less-explored actions, preventing premature convergence to suboptimal choices. 3. Mathematical Convenience: The exponential function ensures positive, normalized weights and efficient updates, making the algorithm scalable and practical for real-time applications.
> The specific choice of ηt is a decaying lower bound on query probability to encourage exploration at an early stage, to reveal the true label more often at an early stage to differentiate the different policies and models regardless how different they present the agreement on the label. The choice of exact value of ηt is guided by our theoretical analysis. Our empirical results further validate the effectiveness of ηt.
>
>
> > ***Q4:*** "The paper does not discuss the stability of empirical results across multiple runs. Given that the evaluation relies on certain stochastic processes, it would be beneficial to report the variance or standard deviation of the results to provide a clearer picture of the method's reliability."
>
> ***A:*** Thank you for the comments. In our experiments, we indeed ran multiple trials for each dataset. Specifically, for the DRIFT dataset, we conducted 100 trials; for the HIV dataset, we conducted 200 trials; for the VERTEBRAL dataset, we conducted 300 trials; for the CIFAR10 dataset, we conducted 10 trials; and for the COVTYPE dataset, we conducted 6 trials. We visualized the results in each plot with a 90% confidence interval (if we approximate the outcomes over multiple trials with a Gaussian distribution, then the confidence interval is approximately proportional to the standard deviation). We will clarify this in the revised manuscript.

---

> ### Author Response · Authors · 2024-08-07
>
> > ***Q5:*** "How does CAMS handle scenarios where the set of pre-trained classifiers is not fixed or continuously updated?"
>
> ***A:*** CAMS is primarily designed for pre-trained classifiers. However, it also provides performance guarantees for scenarios where classifiers are not fixed, continuously updated, or exhibit unexpected or adversarial behavior. We have both algorithms and theoretical bounds in adversarial settings to ensure performance in these situations.
>
> > ***Q6:*** "Can more details be provided on the selection and construction of the policy set used in experiments?"
>
> ***A:*** We select and construct policies with the goal of creating a more diversified policy set, incorporating diversity in features, architecture, and behavior. To achieve this, we adopt models with entirely different architectures to learn contextual representations alongside classifier behavior. Ultimately, we combine these policies to form a stronger model selection strategy.
>
> > ***Q7:*** "How does the method perform when applied to tasks beyond classification, such as regression or ranking problems?"
>
> ***A:***  Currently, CAMS only covers classification tasks where the oracle policy provides a label. When the oracle policy provides a regression value or a ranking list, one possible solution is to convert the problem into a classification problem. This could involve marking the true label as the top in the ranking list or identifying the value closest to the true value as the highest.
>
> ---
> We hope our response has addressed your concerns. If you have any further inquiries, please let us know. Thank you!

---

> ### Comment · Area_Chair_utKU · 2024-08-12
> **Please update your review and engage with the authors**
>
> Dear reviewer,
>
> Please provide an update to your review. The authors have provided a quite substantial rebuttal. Please acknowledge that you have read the rebuttal, and please post any questions that you may still have. Also clarify if you want to adjust your score.
>
> Many thanks, Your AC.

---

> ### Comment · Area_Chair_utKU · 2024-08-13
> **Please respond**
>
> Dear reviewer,
>
> Thanks again for your thoughtful review. As this paper is a bit borderline, I would really like to know if the rebuttal had any affect on your review.
>
> Therefore please provide an update to your review and acknowledge that you have read the rebuttal and clarify if you want to adjust your score.
>
> Many thanks, Your AC.

---

### Official Review · Reviewer_PzjT · 2024-07-14

**Soundness:** 3
**Presentation:** 3
**Contribution:** 2
**Rating:** 5
**Confidence:** 3

**Summary:**

This paper proposes a Contextual Active Model Selection (CAMS) method for addressing the problem in the online setting by selecting the optimal pre-trained model for given data points while minimizing labeling costs. CAMS utilizes contextual information to make informed model selection decisions and employs an adaptive query strategy to determine when to request labels, thereby reducing overall labeling efforts.

**Strengths:**

- This paper provides very rigorous theoretical guarantees, demonstrating the algorithm's effectiveness through regret and query complexity bounds.

- Compared with existing active learning approaches, CAMS's ability to efficiently handle various data distributions/scenarios and contexts makes it particularly useful for real-life applications.

**Weaknesses:**

- The effectiveness of CAMS heavily relies on the pre-trained models. CAMS may not perform well if the pre-trained models are diverse or representative enough or have a large domain gap with the target task.

- The theoretical guarantee of regret bounds and query complexity in this manuscript, are derived under strong assumptions like stochastic, adversarial settings.

- The authors use the cumulative loss as the evaluation metric, it is indeed an overall measure of performance but the normal evaluation metrics like accuracy, precision, and recall should also be considered.

**Questions:**

The author should carefully check if typos exist in the manuscript, e.g., the caption of Table 3, "Eq. (??)".

**Limitations:**

The authors have adequately addressed the limitations and potential negative societal impact of their work.

---

> ### Author Rebuttal · Authors · 2024-08-07
>
> Thank you for your thorough review of our work! We greatly appreciate your recognition of the rigorous theoretical guarantees provided in this paper, both in terms of regret and query complexity bounds. We are also thankful for your acknowledgment of CAMS as particularly useful for real-life applications, filling the gap in current active learning settings.  Below, please find our detailed responses to your questions.
>
> > ***Q1:*** "The effectiveness of CAMS heavily relies on the pre-trained models. CAMS may not perform well if the pre-trained models are diverse or representative enough or have a large domain gap with the target task."
>
> ***A:*** CAMS will generally benefit from the diversity of pretrained models; however, when all the models present a large domain gap with the target task, CAMS will be limited by the combinational optimal performance of current models and policies. To address such concern in the worst case scenario, our adversarial setting provides a worst case performance guarantee when all models have a large domain gap with the target task.
>
> > ***Q2:*** "The theoretical guarantee of regret bounds and query complexity in this manuscript, are derived under strong assumptions like stochastic, adversarial settings."
>
> ***A:*** Thanks for raising the question! Yes, we provide theoretical guarantees of regret bounds and query complexity in both stochastic and adversarial settings. These settings are more like a general setting, not considered to be a strong assumption setting.
> * *Stochastic Settings* deal with inherent randomness and probabilistic outcomes, suitable for applications involving natural variability and uncertainty. These models are essential in various real-world applications due to the inherent randomness in many systems. Such as economic forecasting, chemical reaction, and biological population dynamics.
> * *Adversarial Settings* involve deliberate attempts to disrupt or manipulate systems, requiring strategies to anticipate and counteract adversarial actions. This paradigm is often used in contexts where there is a deliberate effort to deceive or hinder operations. Real-World applications of adversarial settings: such as recommendation systems etc.
>
>
> > ***Q3:*** "The authors use the cumulative loss as the evaluation metric, it is indeed an overall measure of performance but the normal evaluation metrics like accuracy, precision, and recall should also be considered."
>
> ***A:*** Thanks for the question! In this work, we follow the online learning literature to report the cumulative loss (as a proxy of the cumulative regret). As the reviewer rightfully suggested, employing other metrics could convey additional information. We would like to note that the cumulative loss is linear to Accuracy, i.e., Cumulative loss = T*(1-Accuracy), where T is the total number of queries seen by the algorithm. We further considered additional metrics such as the relative cumulative loss (Fig.4(a)), and query complexity (Fig.3(b)). We will make this clear in the revision.
>
> > ***Q4:*** "The author should carefully check if typos exist in the manuscript, e.g., the caption of Table 3, Eq. (??)."
>
> ***A:*** Thank you for pointing out the typo. We have fixed it and will update it in the camera-ready version.
>
>
> ---
> We hope our response has addressed your concerns. If you have any further inquiries, please let us know. Thank you!
>
> ----

---

> > ### Comment · Reviewer_PzjT · 2024-08-11
> > **response**
> >
> > Thank you for your response, I decide to keep my attitude towards borderline accept.

---

### Official Review · Reviewer_jCcC · 2024-07-22

**Soundness:** 4
**Presentation:** 3
**Contribution:** 4
**Rating:** 7
**Confidence:** 3

**Summary:**

The paper proposes an online active model selection strategy where at each round the learner receives an unlabeled data point as a context to adaptively select the best model to predict while limiting the label requests.

**Strengths:**

1.	The paper introduces a model selection procedure that is designed to handle both stochastic and adversarial settings. Apart from that it includes an adaptive query strategy that considers the disagreement among the pre-trained models.

2.	The framework is cost-effective through its adaptive query strategy and performs significantly well compared to all the contextual and non-contextual baselines.

**Weaknesses:**

1.	Mainstream large-scale datasets such as ImageNet, MS COCO etc. will be ideal to validate the all-around performance of the proposed CAMS framework, especially in the query cost and complexity studies.

2.	It will be interesting to check with some of the popular and more recent baselines such as CoreSet[1], BatchBALD[2], BADGE[3] on any of the large datasets mentioned in 1.


[1] Sener, O., & Savarese, S. (2017). Active Learning for Convolutional Neural Networks: A Core-Set Approach. ArXiv. /abs/1708.00489

[2] Andreas Kirsch, Joost van Amersfoort, and Yarin Gal. 2019. BatchBALD: efficient and diverse batch acquisition for deep Bayesian active learning. Proceedings of the 33rd International Conference on Neural Information Processing Systems. Curran Associates Inc., Red Hook, NY, USA, Article 631, 7026–7037.

[3] Ash, J. T., Zhang, C., Krishnamurthy, A., Langford, J., & Agarwal, A. (2019). Deep Batch Active Learning by Diverse, Uncertain Gradient Lower Bounds. ArXiv. /abs/1906.03671

**Questions:**

Please refer to the weaknesses section.

**Limitations:**

The paper is somewhat difficult to read due to its proposal’s components explained in disconnected sections. The framework is not exactly flexible to be applied to regression or segmentation problems.  As reported, this is primarily applicable to classification problems.

---

> ### Author Rebuttal · Authors · 2024-08-07
>
> Thank you for your detailed review of our work! Below, you will find our detailed responses to your questions.
>
> ---
> >***Q1:*** "Mainstream large-scale datasets such as ImageNet, MS COCO etc. will be ideal to validate the all-around performance of the proposed CAMS framework, especially in the query cost and complexity studies."
>
> ***A:*** Thanks for raising the question! To address your concerns, we conducted additional experiments using a more recent and complex large-scale dataset, the ImageNet dataset with 1000 categories. We also incorporated six more recent pretrained models, including CLIP, Inception V4, VGG19, and PNASNet. In Fig. 16 of the attached global response PDF, we present our studies on cost-effective query experiments with the ImageNet dataset using these newer pretrained models. The results are consistent with our previous findings, demonstrating that CAMS outperforms all baselines. CAMS not only shows significant superiority over both contextual and non-contextual baselines but also achieves the lowest label query cost compared to existing baselines and the current state-of-the-art, ModelPicker [1].
> In addition to ImageNet, we also conducted a cost-effective experiment on a relatively large dataset, Covtype (580K instances), shown in Fig. 3(e) and Fig. 4 in the main paper.
>
> [1] Online Active Model Selection for Pre-trained Classifiers, AISTATS 2021
> >***Q2:*** "It will be interesting to check with some of the popular and more recent baselines such as CoreSet[1], BatchBALD[2], BADGE[3] on any of the large datasets mentioned in 1."
>
> ***A:*** Please note that CAMS (ModelPicker and the other AL criteria adopted in the paper) is developed under the streaming setting, where data arrives sequentially or online, and the model decides which label to query. Although one can make multiple passes over the data stream, the decision of whether to query a label is dependent only on the collection of existing models (or hypotheses).
>
> On the other hand, CoreSet, BADGE, and (Batch-)BALD are all designed as pool-based active learning baselines (where CoreSet represents a diversity sampling strategy, BatchBALD being uncertainty sampling, and BADGE representation a combination of both). The active query strategy that CAMS relies on can be viewed as a customized variant of entropy sampling, applied to the streaming setting. This in principle aligns with BALD, with a key difference in that BALD is designed for the pool-based setting as a greedy uncertainty sampling strategy. However, one cannot readily apply diversity sampling under our (streaming) problem setup.
>
>
> | **Active Learning Setting \ Algorithms** | **Coreset (2017)** | **Batch-BALD (2019)** | **BADGE (2019); VAAL (2019); ClusterMargin (2021)** | **BALANCE (2023); GLISTER (2020)** | **VeSSAL (2023)** | **Model Picker (2021)** | **CAMS** |
> |-------------------------------------------|-------------|----------------|-------------------------------|----------------------|------------|------------------|----------|
> | *Streaming, sequential*                   | ×           | ×              | ×                             | ×                    | ×          | ✔️               | ✔️       |
> | *Streaming, batch*                        | ×           | ×              | ×                             | ×                    | ✔️         | ×                | ×        |
> | *Pool-based, batch*                       | ✔️          | ✔️             | ✔️                            | ✔️                   | ×          | ×                | ×        |
>
>
> We also will modify section 2(related works) to address this concern.
>
> ---
> We hope our response has addressed your concerns. If you have any further inquiries, please let us know. Thank you!

---

> > ### Comment · Reviewer_jCcC · 2024-08-14
> >
> > Thanks for your response, I maintain my initial score.

---

### Author Rebuttal · Authors · 2024-08-07

We would like to thank all reviewers for their effort in assessing our work and for their helpful comments and questions.

We will respond separately to each reviewer concerning their individual questions. However, we would like to address one overarching theme from the reviews upfront: the performance and scalability of CAMS on large datasets and our choice of baselines. Our response is as follows:

1. **Scalability and robustness of CAMS**: To further demonstrate the scalability and robustness of CAMS, we conducted additional experiments on ImageNet and incorporated six more recent pretrained models, including CLIP, Inception V4, VGG19, and PNASNet. The results were consistent with those reported in our original submission, and the new results are provided in the attached PDF.

2. **Problem setting and baseline selection**: As shown in Table 1, the problem setting of contextual active model selection significantly differs from classical contextual bandits and active learning problems. Therefore, we focused on demonstrating that a novel combination of classical algorithmic components, namely EXP4 (for contextual bandits) and uncertainty sampling (for streaming queries), can elegantly solve this new problem. This problem is practically relevant, as pointed out by several reviewers.

3. **Comparison against recent studies in active learning**: In addition to Table 1 of our original submission, which showcases the novelty of our problem, we have added another table in the attached PDF to highlight the differences between the active model selection problem and recent studies that explicitly focus on active learning. We hope this justifies our choice of baselines as a fair selection for a well-rounded evaluation framework.

We will gladly incorporate all of these aspects in our revised manuscript.

---

### Decision · Program_Chairs · 2024-09-25

**Decision:**

Accept (poster)

**Comment:**

This paper poses and solves an important and interesting practical problem well, and we think this paper will be impactful. While the reviewers raised a number of interesting questions, there were no fundamental concerns, and most of the questions really just point to possible extensions, such as how to train good models given the proposed process, or how to adapt to changing models.